# Dual Regularized Optimal Transport

## Abstract

In this paper, we present a new formulation of unbalanced optimal transport called Dual Regularized Optimal Transport (DROT). We argue that regularizing the dual formulation of optimal transport results in a version of unbalanced optimal transport that leads to sparse solutions and that gives us control over mass creation and destruction. We build intuition behind such control and present theoretical properties of the solutions to DROT. We demonstrate that due to recent advances in optimization techniques, we can feasibly solve such a formulation and present extensive experimental evidence for this formulation and its solution.

## 1 Introduction

Optimal transport is a ubiquitous problem in areas ranging from economics and the allocation of resources to Riemannian geometry and measure theory. The motivation for and description of the basic problem arises from transporting objects from one set of locations to the another using a minimal cost transportation plan. Over the past century, but especially the last three decades, considerable work has been done to understand the geometry of the problem and its various formulations. Many different variations of the problem have been posed and algorithmic approaches have been developed to solve these variants. Most importantly for our work, there has also been great interest and activity in applying optimal transport to machine learning, computer vision, and domain transfer tasks. Optimal transport in the setting of machine learning tasks is the starting point of this paper.

### 1.1 Background

There are several versions of the optimal transport problem that we use to motivate our formulation. The original version is that of Monge which however, has some drawbacks (namely, the transport map must be a function) and, for this reason, we begin with its natural generalization, the Monge-Kantorovich problem.

**Problem 1.** *Given two probability spaces $(\mathcal{X}, \mu)$ and $(\mathcal{Y}, \nu)$, and a cost function $c : \mathcal{X} \times \mathcal{Y} \to \mathbb{R}^+$, the Monge-Kantorovich Optimal Transport seeks a joint probability measure $\pi$ on $\mathcal{X} \times \mathcal{Y}$ that minimizes $\int_{\mathcal{X} \times \mathcal{Y}} c(x, y) d\pi(x, y)$, subject to the constraints that the pushforward of the marginals are consistent with the inputs, $\mathcal{P}^{\mathcal{X}}_{\#} \pi = \mu$ and $\mathcal{P}^{\mathcal{Y}}_{\#} \pi = \nu$.*

In a finite discrete setting this problem can be formulated as a linear program (see Problem 2) which, unfortunately, is challenging to solve computationally but which does guarantee sparse solutions. Two predominant methods are combinatorial flows Bertsekas & Castanon (1989); Gabow (1985); Duff & Koster (2001) and PDE based solvers Benamou & Brenier (2000). None of these methods, however, scales well. As a result, there are many alternative formulations of the OT problem that are easier to solve, including those formulation types that include regularizing the primal objective function (see, for example, Cuturi (2013); Essid & Solomon (2017); Blondel et al. (2018); Lorenz et al. (2019) ) with or without relaxed constraints. These variants are referred to as regularized optimal transport. There is a second class of formulations called unbalanced optimal transport (see for example Liero et al. (2017); Chizat et al. (2016); Blondel et al. (2018). There are a number of proposed efficient algorithms to solve these various formulations, including Seguy et al. (2018); Schmitzer (2019); Solomon et al. (2015); Frogner et al. (2015); Benamou et al. (2015); Genevay et al. (2016); Alaya et al. (2019).

### 1.1.1 Our Contribution

In this paper, we present a new formulation of optimal transport that regularizes the dual problem without relaxing the dual constraints. We refer to this formulation as Dual Regularized Optimal Transport or DROT. We show that this problem has a number of both theoretical and algorithmic properties that are desirable. Specifically,

1. the dual of DROT is a form of unbalanced optimal transport whose solution leads to sparse solutions to the optimal transport problem;
2. the solutions to DROT are good approximations to the solution of the Monge-Kantorovich problem;
3. **with the appropriate choice of the dual regularizer, unlike other optimal transport formulations, we can easily control the level of mass creation versus the level of mass destruction and**
4. DROT can be solved efficiently via PROJECT AND FORGET, a general optimization method developed in Gilbert & Sonthalia (2020).

## 2 Preliminaries

For all of our algorithmic discussions, we work in a finite, discrete setting. Let $\Delta^n$ denote the $n-1$ dimensional probability simplex. Then, $(\Delta^m, \boldsymbol{a})$ and $(\Delta^n, \boldsymbol{b})$ denote two finite probability spaces and we denote by $\boldsymbol{P}$ the joint distribution on $\Delta^m \times \Delta^n$. Note that $\boldsymbol{P}$ can be represented by an $m \times n$ matrix. The cost function we denote by an $m \times n$ matrix $\boldsymbol{C}$. The vector of all ones of length $m$ is denoted $\mathbf{1}_m$. The Frobenius dot product of two matrices $\boldsymbol{A}, \boldsymbol{B}$ we denote by $\langle \boldsymbol{A}, \boldsymbol{B} \rangle$. For some problem formulations and in an abuse of notation, the distributions $\boldsymbol{a}$ and $\boldsymbol{b}$ on their respective spaces need not have the same total mass (i.e., they are not strictly probability measures). Finally, given a convex function $\phi$, we denote its convex conjugate by $\phi^*$.

### 2.1 Background Problem Formulations

In a finite discrete setting the Monge-Kantorovich OT problem can be formulated as the following linear problem.

**Problem 2.** *Given two probability spaces $(\Delta^m, \boldsymbol{a})$ and $(\Delta^n, \boldsymbol{b})$ and a cost function $\boldsymbol{C}$, we seek the mass transportation map of minimal cost that is consistent with the input distributions:*

$$\mathrm{OT}(\boldsymbol{a}, \boldsymbol{b}) = \min \langle \boldsymbol{C}, \boldsymbol{P} \rangle$$
$$\textit{subject to: } \boldsymbol{a} = \boldsymbol{P}\mathbf{1}_m, \boldsymbol{b} = \boldsymbol{P}^T\mathbf{1}_n, \boldsymbol{P} \geq 0. \tag{1}$$

One important feature of the solution to Problem 2 is that it is sparse. Specifically, at most $n + m - 1$ entries of $\boldsymbol{P}$ are non-zero Brualdi (2006) which means that for applications in machine learning and image processing, the solutions are "interpretable" and they have efficient implementations.

In what follows, we sketch the problem formulation types that include regularizing the primal objective function with and without relaxed constraints.

### 2.1.1 Regularized and Unbalanced Optimal Transport

In the first formulation variant Regularized Optimal Transport (or ROT), we use an entropic regularizer without relaxing the constraints. Cuturi Cuturi (2013) shows that by adding an entropic regularizer, the ROT problem can be solved quickly with the Sinkhorn matrix scaling algorithm.

$$\mathrm{ROT}(\boldsymbol{a}, \boldsymbol{b}) = \min \langle \boldsymbol{C}, \boldsymbol{P} \rangle + \gamma \sum_{i,j} \boldsymbol{P}_{ij} \log(\boldsymbol{P}_{ij})$$
$$\textit{subject to: } \boldsymbol{a} = \boldsymbol{P}\mathbf{1}_m, \boldsymbol{b} = \boldsymbol{P}^T\mathbf{1}_n, \boldsymbol{P} \geq 0. \tag{2}$$

This formulation has proven to be extremely useful in practice despite the loss in sparsity of the solution which smooths the transportation plan.

A second natural regularizer is the quadratic function. Essid & Solomon (2017); Blondel et al. (2018); Lorenz et al. (2019) study this variant and show experimentally that the solutions are sparse. Generalizing further, Dessein et al. (2018b) use Bregman functions, a natural extension of Benamou et al. (2015).

A second main formulation variant, Unbalanced Optimal Transport (or UOT), maintains the regularized primal objective function but relaxes the constraints on the marginal distributions. In a variety of applications, the input distributions are *not* probability measures and they have *different* total mass. As a result, Chizat et al. (2015); Liero et al. (2017) formulate transport between densities with different masses, or unbalanced optimal transport. In this variant, we relax the constraint that marginals of the transport must match the given marginals and instead penalize the deviation from the marginals. Similar to Cuturi (2013), Liero et al. (2017) use entropy based divergences, such as the KL divergence, as the penalty function. Chizat et al. (2016) present matrix scaling algorithms for UOT.

$$\text{UOT}(\boldsymbol{a}, \boldsymbol{b}) = \min \langle \boldsymbol{C}, \boldsymbol{P} \rangle + \gamma_1 \sum_{i,j} \boldsymbol{P}_{ij} \log(\boldsymbol{P}_{ij}) + \gamma_2 KL(\boldsymbol{P}\mathbf{1}_m, \boldsymbol{a}) + \gamma_3 KL(\boldsymbol{P}^T\mathbf{1}_n, \boldsymbol{b}). \tag{3}$$

Blondel et al. (2018) consider UOT with quadratic penalty terms and also considers an asymmetric version of the problem in which only one marginal constraint has been relaxed. The Monge version of the problem also has a relaxation that is similar to the unbalanced version of the Monge-Kantorovich problem Yang & Uhler (2019).

The main drawback with the current formulations of unbalanced optimal transport is that it is unclear how the solution methods balance creation, destruction, and transportion of mass. These formulations give us control over mass creation and destruction versus transportion, by increasing or decreasing the penalty, but we do not have control over the degree of creation versus that of destruction.

There is a version of the unbalanced problem known as partial optimal transport Caffarelli & McCann (2010); Figalli (2010) that allows for some control. It does require that the input distributions dominate the learned marginal distributions. As we shall show, our setting not only captures such optimal transport, but also captures the mass creation version as well. That is, the learned marginals dominate the input distributions.

## 2.2 Dual regularized optimal transport (DROT)

We devise a new formulation of OT via dual regularization. We add a regularizer term to the dual objective function so that it is strictly concave but we do *not* relax the dual constraints. This may be interpreted as adding a strictly convex regularizer to the primal problem and relaxing the primal constraints, leading to an unbalanced optimal transport problem. We state the discrete version of the problem and note there is a natural continuous version which we do not state.

**Problem 3.** *Given $\boldsymbol{a}$ and $\boldsymbol{b}$ two vectors of length $m$ and $n$ respectively (representing two distributions on $m$ and $n$ points), an $m \times n$ cost matrix $\boldsymbol{C}$, two strictly convex function $\varphi$ and $\phi$, and a regularization parameter $\gamma$, find vectors $\boldsymbol{f}$ and $\boldsymbol{g}$ that maximize*

$$\text{DROT}(\boldsymbol{a}, \boldsymbol{b}) = \max \langle \boldsymbol{f}, \boldsymbol{a} \rangle + \langle \boldsymbol{g}, \boldsymbol{b} \rangle - \frac{1}{\gamma}(\phi(\boldsymbol{f}) + \varphi(\boldsymbol{g}))$$

*subject to: $\boldsymbol{f}_i + \boldsymbol{g}_j \leq \boldsymbol{C}_{i,j}.$* \tag{4}

Let us consider the interpretation of this formulation. We begin with that of Peyré & Cuturi (2018). Suppose we have $n$ warehouses and $m$ stores. Let $\boldsymbol{a}$ be the vector whose $i$th component is the number of items in warehouse $i$ and $\boldsymbol{b}$ be the $m$ dimensional vector for the demand of each store. Let $\boldsymbol{C}$ be the cost to transport items from warehouses to stores. Next, suppose we are an external shipper; we charge $\boldsymbol{f}_i$ to pick up goods from warehouse $i$ regardless of where they are delivered and $\boldsymbol{g}_j$ to deliver goods to store $j$ regardless of the originating warehouse. We want to maximize our income which is given by $\langle \boldsymbol{f}, \boldsymbol{a} \rangle + \langle \boldsymbol{g}, \boldsymbol{b} \rangle$ but our prices must satisfy $\boldsymbol{f}_i + \boldsymbol{g}_j \leq \boldsymbol{C}_{ij}$, some cost constraint. The addition of the regularizer in the objective function, therefore, regularizes the prices we can charge. This is in contrast with the formulation developed in Liero et al. (2017) which penalizes the divergence from the input distribution. In many applications, such as domain transfer, color transfer, and economics, regularizing prices (i.e., how profitable is it to transfer both to and from a

certain data point) is more natural. For example, we may want to regularize prices and see how this affect this the distributions $\boldsymbol{a}, \boldsymbol{b}$, representing demand and supply.

Using this interpretation, we note the following intuition. If we regularize the problem so that $\boldsymbol{f}, \boldsymbol{g}$ are large, then we as shippers are making a large profit. Hence, we want to ship as much as possible. Thus, such regularizers would lead to solutions that create mass. On the other hand, if we regularize so that $\boldsymbol{f}, \boldsymbol{g}$ are small, negative even, then we as shippers do not want to ship goods. Hence, such regularizers lead to solutions that destroy mass.

## 3 Theoretical analysis

In this section, we detail the theoretical analysis of the DROT problem formulation. We begin with an analysis of the features of the solutions. We then discuss the choice of regularizer. We end with a discussion of an algorithmic method for solving Problem 4, PROJECT AND FORGET, a general method developed in Gilbert & Sonthalia (2020).

### 3.1 Solution properties

In this section, we analyze the properties of the solutions to the DROT problem. This analysis includes the relation between the solution to the DROT Problem 4 and that of other OT formulations (i.e., the approximation quality of the solution), how the solutions depend on the regularization parameter, and finally, what the trade-offs are in the creation and destruction of mass.

**Definition 1.** *Let $f : \mathbb{R}^n \to \mathbb{R}$. We say $f$ is positive co-finite if for all $x \geq 0$, $f(rx)/r \to \infty$ as $r \to \infty$. Similarly, $f$ is negative co-finite if for all $x \leq 0$, $f(rx)/r \to \infty$ as $r \to \infty$. A function is co-finite if it is both positive and negative co-finite.*

**Definition 2.** *Given a fucntion $h : \mathbb{R}^n \to \mathbb{R}$, the convex conjugate $h^* : \mathbb{R}^n \to \mathbb{R}$ is defined as follows*

$$h^*(x^*) = \sup_{x \in \mathbb{R}^n} x^T x^* - h(x).$$

**Theorem 1.** *If we add the assumption that $\phi, \varphi$ are co-finite Bregman functions to our hypotheses for Problem 4, then the following problem is the dual problem to DROT($\boldsymbol{a}, \boldsymbol{b}$). Furthermore, strong duality holds.*

$$\min \langle \boldsymbol{C}, \boldsymbol{P} \rangle + \frac{\phi^*\left(\gamma(\boldsymbol{a} - \boldsymbol{P}\mathbf{1}_m)\right)}{\gamma} + \frac{\varphi^*\left(\gamma(\boldsymbol{b} - \boldsymbol{P}^T\mathbf{1}_n)\right)}{\gamma} \tag{5}$$

$$\text{subject to: } \forall i \in [n], \forall j \in [m], \quad \boldsymbol{P}_{ij} \geq 0.$$

*If we only have the assumption that $\phi$ (and similarly for $\varphi$) is positively (negatively) co-finite, then we must add the constraint $\boldsymbol{a} - \boldsymbol{P}\mathbf{1} > 0$ ($\boldsymbol{a} - \boldsymbol{P}\mathbf{1} < 0$).*

*Proof.* Since Bregman functions are strictly convex and we have linear inequality constraints, it is easy to see that DROT($\boldsymbol{a}, \boldsymbol{b}$) is a convex program. Furthermore, strong duality holds if Slater's condition Slater (2014) holds. Specifically, given $\boldsymbol{C}$, we need to show the existence of an $\boldsymbol{f}$ and $\boldsymbol{g}$ such that for all $i, j$ we have that $\boldsymbol{f}_i + \boldsymbol{g}_j < \boldsymbol{C}_{ij}$. To do so, set

$$\boldsymbol{f} = -\|\boldsymbol{C}\|_\infty \mathbf{1}_n \text{ and } \boldsymbol{g} = -\|\boldsymbol{C}\|_\infty \mathbf{1}_m.$$

Thus, we have strong duality.

Let us now compute the dual of the problem. To do so, let $\boldsymbol{P}$ be the dual variables and obtain the Lagrangian $L(\boldsymbol{f}, \boldsymbol{g}, \boldsymbol{P})$:

$$L(\boldsymbol{f}, \boldsymbol{g}, \boldsymbol{P}) = \frac{1}{\gamma}\phi(\boldsymbol{f}) + \frac{1}{\gamma}\varphi(\boldsymbol{g}) - \boldsymbol{f}^T\boldsymbol{a} - \boldsymbol{g}^T\boldsymbol{b} + \langle \boldsymbol{P}, \boldsymbol{f}\mathbf{1}_m^T + \mathbf{1}_n\boldsymbol{g}^T - \boldsymbol{C} \rangle. \tag{6}$$

Now we know that the dual problem is given by

$$\max_{\boldsymbol{P}_{ij} \geq 0} \min_{\boldsymbol{f}, \boldsymbol{g}} L(\boldsymbol{f}, \boldsymbol{g}, \boldsymbol{P}). \tag{7}$$

Let us do some simplifications to get this into the standard form. We first note that the Lagrangian $L$ can be rewritten as

$$L(\boldsymbol{f}, \boldsymbol{g}, \boldsymbol{P}) = \frac{1}{\gamma}\phi(\boldsymbol{f}) + \frac{1}{\gamma}\varphi(\boldsymbol{g}) - \langle \boldsymbol{f}, \boldsymbol{a} - \boldsymbol{P}\mathbf{1}_n \rangle - \langle \boldsymbol{g}, \boldsymbol{b} - \boldsymbol{P}^T\mathbf{1}_m \rangle - \langle \boldsymbol{P}, \boldsymbol{C} \rangle. \tag{8}$$

Now, for fixed $\boldsymbol{P}$ consider the function

$$F(\boldsymbol{f}) = \frac{1}{\gamma}\phi(\boldsymbol{f}) - \langle \boldsymbol{f}, \boldsymbol{a} - \boldsymbol{P}\mathbf{1}_m \rangle.$$

Due to the strict convexity and co-finiteness of $\phi$, we have that $F$ is a strictly convex function. and has a unique stationary point that corresponds to its global minimum $\boldsymbol{f}^*$. We can solve for this as follows. For the case when we have positive co-finiteness only, we need $\boldsymbol{a} - \boldsymbol{P} < 0$ for $F$ to have a stationary point. Note if these conditions are not satisfied then the value of $L(\boldsymbol{f}, \boldsymbol{g}, \boldsymbol{P})$ is negative infinity, however if it is satisfied then it is a finite number. Thus, since we have the outer maximization, this is equivalent to adding the constraint.

$$0 = \nabla F(\boldsymbol{f}^*) = \frac{1}{\gamma}\nabla\phi(\boldsymbol{f}^*) - \boldsymbol{a} + \boldsymbol{P}\mathbf{1}_m.$$

Thus, we have that

$$\frac{1}{\gamma}\nabla\phi(\boldsymbol{f}^*) = \boldsymbol{a} - \boldsymbol{P}\mathbf{1}_m.$$

Now from Bauschke & Borwein (1998), if we can show that $\phi$ is *essentially strictly convex* then due to $\phi$ being co-finite, we have that $\nabla\phi^*\nabla\phi(\boldsymbol{f}) = \boldsymbol{f}$.

**Lemma 1.** *If $\phi$ is a Bregman function, then $\phi$ is* essentially strictly convex

*Proof.* From Rockafellar (1970), we know that a function $\phi$ is *essentially strictly convex* if for all convex $S \subset \{x : \nabla\phi(x) \neq 0\} =: \text{dom}(\partial\phi)$, $\phi$ is strictly convex on $S$. From Rockafellar (1970), we also know that $\text{dom}(\partial\phi) \subset \text{dom}\phi$. Thus, since Bregman functions are strictly convex, we have that $\phi$ is *essentially strictly convex*. $\square$

Hence via Lemma 1, we have that

$$\boldsymbol{f}* = \nabla\phi^*\left(\gamma(\boldsymbol{a} - \boldsymbol{P}\mathbf{1}_m)\right)$$

Performing a similar calculation for $\boldsymbol{g}$ and substituting into Equation 7, we get the following equation for dual.

$$\max_{\boldsymbol{P}_{ij} \geq 0} -\langle C, P \rangle + \frac{1}{\gamma}\phi(\nabla\phi^*(\gamma(\boldsymbol{a} - \boldsymbol{P}\mathbf{1}_m))) - \langle \nabla\phi^*(\gamma(\boldsymbol{a} - \boldsymbol{P}\mathbf{1}_m)), \boldsymbol{a} - \boldsymbol{P}\mathbf{1}_n \rangle$$

$$+ \frac{1}{\gamma}\varphi(\nabla\varphi^*(\gamma(\boldsymbol{b} - \boldsymbol{P}^T\mathbf{1}_m))) - \langle \nabla\varphi^*(\gamma(\boldsymbol{b} - \boldsymbol{P}^T\mathbf{1}_m)), \boldsymbol{b} - \boldsymbol{P}^T\mathbf{1}_n \rangle$$

To simplify this, Amari (2016) tells us that

$$\psi^*\left(\nabla\psi(x)\right) = x^T\nabla\psi(x) - \psi(x) \tag{9}$$

From Rockafellar (1970), we know that $\phi^{**} = cl(conv(\phi))$. However, since $\phi$ is closed and convex, we have that $\phi^{**} = \phi$. Additionally, since we also have that $\phi^*$ is closed and convex Rockafellar (1970), we also have that $\phi^{***} = \phi^*$. Thus, we have that

$$\frac{1}{\gamma}\phi(\nabla\phi^*(\gamma(\boldsymbol{a} - \boldsymbol{P}\mathbf{1}_m))) = \langle \boldsymbol{a} - \boldsymbol{P}\mathbf{1}_m, \nabla\phi^*(\gamma(\boldsymbol{a} - \boldsymbol{P}\mathbf{1}_m)) \rangle$$

$$- \frac{1}{\gamma}\phi^*(\gamma(\boldsymbol{a} - \boldsymbol{P}\mathbf{1}_m))$$

Substituting back, we get that dual of $\mathrm{DROT}(\boldsymbol{a}, \boldsymbol{b})$ is given by

$$\text{Minimize:} \quad \langle \boldsymbol{C}, \boldsymbol{P} \rangle + \frac{\phi^*(\gamma(\boldsymbol{a} - \boldsymbol{P}\mathbf{1}_m))}{\gamma} + \frac{\varphi^*(\gamma(\boldsymbol{b} - \boldsymbol{P}^T\mathbf{1}_n))}{\gamma}$$

$$\text{Subject to:} \quad \forall i \in [n], \forall j \in [m], \quad \boldsymbol{P}_{ij} \geq 0$$

$\square$

**Remark 1.** *Our proof of strong duality, as written, does not hold for the entropic regularizer. For this regularizer we need to add the assumption that $\boldsymbol{C}_{ij} > 0$ for all $i, j$. With such an assumption, setting $\boldsymbol{f}, \boldsymbol{g} = 0$ satisfies Slater's condition[1]. We also need to add the constraint that $\boldsymbol{f}, \boldsymbol{g} \geq 0$ so, in the dual formulation, we add the dual variables $\boldsymbol{c}_1, \boldsymbol{c}_2$ that correspond to these constraints.*

Theorem 1 shows us the dual formulation of DROT resembles unbalanced optimal transport problems from Chizat et al. (2015); Liero et al. (2017), but with different types of penalty functions on the transport map. Indeed, if we set $\phi$ and $\varphi$ to be quadratic regularizers, then Theorem 1 shows that the dual DROT formulation and a formulation in Blondel et al. (2018) are equivalent.

Furthermore, note that if $\phi, \varphi$ are positive co-finite functions, then DROT necessarily destroys mass. On the other hand, if $\phi, \varphi$ are negative co-finite function, then DROT necessarily creates mass. This matches our intuition exactly. In the objective function for DROT, the regularizer term is $\phi(\boldsymbol{f}) + \varphi(\boldsymbol{g})$ which we seek to minimize. For positive co-finite functions, we do so when both $\boldsymbol{f}$ and $\boldsymbol{g}$ are highly negative. Using the shipping interpretation of the dual problem, $\boldsymbol{f}$ and $\boldsymbol{g}$ represent the prices we charge to ship and a negative price means that we, as shippers, *pay* to do the shipping! Such incentives result in *not* shipping goods or, more abstractly, destroying mass. On the other hand, for negatively co-finite functions, we minimize the objective function when $\boldsymbol{f}, \boldsymbol{g}$ are both highly positive; that is, we are incentivized to ship more goods, or to create mass.

We note that for the dual DROT formulation, it is not necessary that $\phi^*, \varphi^*$ attain their minima at 0 (the minimum is attained at 0 if and only if $\phi, \varphi$ attain their minima at 0) and, under such conditions, the regularizers actually encourage some deviation from the marginals $\boldsymbol{a}, \boldsymbol{b}$; thus, encouraging the creation or destruction of mass. Note we could also introduce similar incentives in other variants, but they appear naturally in this variant.

The next proposition quantifies how far the solution to DROT is from that of the Monge-Kantorovich formulation.

**Proposition 1.** *Suppose $\phi, \psi$ are bounded from below. Let $\boldsymbol{P}^*, \boldsymbol{f}^*, \boldsymbol{g}^*$ be the optimal solutions, primal and dual, to the Monge-Kantorovich formulation (Problem 2) and let $\boldsymbol{P}^*_{\phi,\varphi}, \boldsymbol{f}^*_{\phi,\varphi}, \boldsymbol{g}^*_{\phi,\varphi}$ be the optimal solutions to DROT, Problem 4. Then the following are true.*

1. *The difference between the value of the DROT objective and that of the Monge-Kantorovich formulation is upper and lower bounded by*

$$\phi\left(\boldsymbol{f}^*_{\phi,\varphi}\right) + \varphi(\boldsymbol{g}^*_{\phi,\varphi}) \leq \gamma(OT(\boldsymbol{a}, \boldsymbol{b}) - DROT(\boldsymbol{a}, \boldsymbol{b})) \leq \phi(\boldsymbol{f}^*) + \varphi(\boldsymbol{g}^*).$$

2. *We can estimate the quality of the approximation (as a function of the regularizers $\phi$ and $\varphi$) as*

$$\gamma\langle \boldsymbol{C}, \boldsymbol{P}^* - \boldsymbol{P}^*_{\phi,\varphi} \rangle \leq \phi(\boldsymbol{f}^*) + \varphi(\boldsymbol{g}^*) + \phi^*(\gamma(\boldsymbol{a} - \boldsymbol{P}^*_{\phi,\varphi}\mathbf{1}_m)) + \varphi^*(\gamma(\boldsymbol{b} - (\boldsymbol{P}^*_{\phi,\varphi})^T\mathbf{1}_n))$$

3. *and*

$$\phi^*(\gamma(\boldsymbol{a} - \boldsymbol{P}^*_{\phi,\varphi}\mathbf{1}_m)) + \varphi^*(\gamma(\boldsymbol{b} - (\boldsymbol{P}^*_{\phi,\varphi})^T\mathbf{1}_n)) \leq \gamma\langle \boldsymbol{C}, \boldsymbol{P}^* - \boldsymbol{P}^*_{\phi,\varphi} \rangle - \phi(\boldsymbol{f}^*_{\phi,\varphi}) - \varphi(\boldsymbol{g}^*_{\phi,\varphi}).$$

*Proof.* Let us first prove the lower bound for part 1. To do this note that since $\boldsymbol{f}^*_{\phi,\varphi}$ and $\boldsymbol{g}^*_{\phi,\varphi}$ satisfy the constraints $\boldsymbol{f}^*_{\phi,\varphi}\mathbf{1}_m^T + \mathbf{1}_n(\boldsymbol{g}^*_{\phi,\varphi})^T \leq \boldsymbol{C}$, we have that

$$\langle \boldsymbol{f}^*_{\phi,\varphi}, \boldsymbol{a} \rangle + \langle \boldsymbol{g}^*_{\phi,\varphi}, \boldsymbol{b} \rangle \leq \langle \boldsymbol{f}^*, \boldsymbol{a} \rangle + \langle \boldsymbol{g}^*, \boldsymbol{b} \rangle = OT(\boldsymbol{a}, \boldsymbol{b})$$

---

[1]For all experiments involving the entropic regularizer, we add a small number to the cost matrix to guarantee that all the costs are positive.

Then subtracting $\frac{1}{\gamma}\phi\left(\boldsymbol{f}^*_{\phi,\varphi}\right) + \frac{1}{\gamma}\varphi\left(\boldsymbol{g}^*_{\phi,\varphi}\right)$ from both sides and rearranging gives us the the lower bound.

For the upper bound, note that $\boldsymbol{f}^*\mathbf{1}_m^T + \mathbf{1}_n(\boldsymbol{g}^*)^T \leq \boldsymbol{C}$, hence we have that

$$\frac{1}{\gamma}\phi(\boldsymbol{f}^*) + \frac{1}{\gamma}\varphi(\boldsymbol{g}^*) - (\boldsymbol{f}^*)^T\boldsymbol{a} - (\boldsymbol{g}^*)^T\boldsymbol{b} \geq -DROT(\boldsymbol{a},\boldsymbol{b}).$$

Thus, rearranging gives us the upper bound.

Now for part 2, we have that

$$\langle \boldsymbol{C},\boldsymbol{P}^*\rangle = \langle \boldsymbol{f}^*,\boldsymbol{a}\rangle + \langle \boldsymbol{g}^*,\boldsymbol{b}\rangle.$$

Then we subtract $(\phi(\boldsymbol{f}^*) + \varphi(\boldsymbol{g}^*))/\gamma$ from both sides to get

$$\langle \boldsymbol{C},\boldsymbol{P}^*\rangle - \frac{1}{\gamma}(\phi(\boldsymbol{f}^*) + \varphi(\boldsymbol{g}^*)) = \langle \boldsymbol{f}^*,\boldsymbol{a}\rangle + \langle \boldsymbol{g}^*,\boldsymbol{b}\rangle - \frac{1}{\gamma}(\phi(\boldsymbol{f}^*) + \varphi(\boldsymbol{g}^*)).$$

Then we have that

$$\langle \boldsymbol{f}^*,\boldsymbol{a}\rangle + \langle \boldsymbol{g}^*,\boldsymbol{b}\rangle - \frac{1}{\gamma}(\phi(\boldsymbol{f}^*) + \varphi(\boldsymbol{g}^*)) \leq DROT(\boldsymbol{a},\boldsymbol{b}).$$

Thus, we get that

$$\langle \boldsymbol{C},\boldsymbol{P}^*\rangle - \frac{1}{\gamma}(\phi(\boldsymbol{f}^*) + \varphi(\boldsymbol{g}^*)) \leq \langle C,\boldsymbol{P}^*_{\phi,\varphi}\rangle + \frac{\phi^*(\gamma(\boldsymbol{a} - \boldsymbol{P}^*_{\phi,\varphi}\mathbf{1}_m))}{\gamma} + \frac{\varphi^*(\gamma(\boldsymbol{b} - (\boldsymbol{P}^*_{\phi,\varphi})^T\mathbf{1}_n))}{\gamma}$$

Rearranging the above equation gives us part 2.

For part 3, note that

$$\langle \boldsymbol{C},\boldsymbol{P}^*\rangle = \langle \boldsymbol{f}^*,\boldsymbol{a}\rangle + \langle \boldsymbol{g}^*,\boldsymbol{b}\rangle \geq \langle \boldsymbol{f}^*_{\phi,\varphi},\boldsymbol{a}\rangle + \langle \boldsymbol{g}^*_{\phi,\varphi},\boldsymbol{b}\rangle.$$

Then we subtract $(\phi(\boldsymbol{f}^*_{\phi,\varphi}) + \varphi(\boldsymbol{g}^*_{\phi,\varphi}))/\gamma$ from both sides to get

$$\langle \boldsymbol{C},\boldsymbol{P}^*\rangle - \frac{1}{\gamma}(\phi(\boldsymbol{f}^*_{\phi,\varphi}) + \varphi(\boldsymbol{g}^*_{\phi,\varphi})) \geq DROT(\boldsymbol{a},\boldsymbol{b})$$

Substituting in the primal objective for DROT and rearranging gives us part 3. $\qquad\square$

These bounds reveal how the various parameters control the problem. Specifically, we can see that error $OT(\boldsymbol{a},\boldsymbol{b}) - DROT(\boldsymbol{a},\boldsymbol{b})$ is $O(\gamma^{-1})$. More interestingly, we see how $\phi,\varphi$ affect the quality of the approximation. Parts 2, 3 of Proposition 1 also give us an interplay between the penalty incurred for not satisfying the marginal constraints and the cost of the transport.

**Corollary 1.** *Suppose $\phi,\psi$ are bounded from below. If $\boldsymbol{P}^*_\gamma$ is the solution to $DROT(\boldsymbol{a},\boldsymbol{b})$ for a given $\gamma$, and $\boldsymbol{P}^*$ is the solution to $OT(\boldsymbol{a},\boldsymbol{b})$ then, $\|\boldsymbol{a} - \boldsymbol{P}^*_\gamma\mathbf{1}_m\|$ and $\|\boldsymbol{b} - (\boldsymbol{P}^*_\gamma)^T\mathbf{1}_n\|$, $OT(\boldsymbol{a},\boldsymbol{b}) - DROT(\boldsymbol{a},\boldsymbol{b})$, and $|\langle \boldsymbol{C},\boldsymbol{P}^* - \boldsymbol{P}^*_\gamma\rangle|$ are all $O(\gamma^{-1})$.*

*Proof.* Note that at the optimal point, by the KKT conditions, we have stationarity. So we have that

$$\frac{1}{\gamma}\nabla\phi(\boldsymbol{f}^*_\gamma) = \boldsymbol{a} - \boldsymbol{P}^*_\gamma\mathbf{1}_m \Rightarrow \|\boldsymbol{a} - \boldsymbol{P}^*_\gamma\mathbf{1}_m\| = \frac{1}{\gamma}\|\nabla\phi(\boldsymbol{f}^*_\gamma)\|.$$

Now due to the convexity of $\phi$, and part 1 of Proposition 1, we have that $\phi(\boldsymbol{f}^*_\gamma)$ is bounded from above. Then again due to the convexity of $\phi$, this implies that $\|\nabla\phi(\boldsymbol{f}^*_\gamma)\|$ is bounded from above, Thus, $\|\boldsymbol{a} - \boldsymbol{P}^*_\gamma\mathbf{1}_m\|$ is $O(\gamma^{-1})$. Similarly, noting that $\phi,\psi$ are bounded from below, due to Proposition 1 part 1, we have that $OT(\boldsymbol{a},\boldsymbol{b}) - DROT(\boldsymbol{a},\boldsymbol{b})$ is $O(\gamma^{-1})$. Finally, since $\|\boldsymbol{a} - \boldsymbol{P}^*_\gamma\mathbf{1}_m\|$ is bounded, we have that $\boldsymbol{a} - \boldsymbol{P}^*_\gamma\mathbf{1}_m$ lives in a bounded set whose diameter is $O(\gamma^{-1})$. Thus, $\phi^*(\gamma(\boldsymbol{a} - \boldsymbol{P}^*_\gamma\mathbf{1}_m))$ is bounded. Thus, using similar reasoning to before and Proposition 1 parts 2 and 3, we have that $|\langle \boldsymbol{C},\boldsymbol{P}^* - \boldsymbol{P}^*_\gamma\rangle|$ is $O(\gamma^{-1})$. $\qquad\square$

Because the sparsity of solutions to OT problems is critical for some applications, the next series of analysis is the study of the support of solutions to DROT.

**Definition 3.** *Given $F : \mathbb{R}^d \times \Theta \to \mathbb{R}$ and $G$ such that for each $\theta \in \Theta$, $G(\theta) \subset \mathbb{R}^d$, we define a parameterized family of optimization problems parameterized by $\theta \in \Theta$ where the function $V$, $V(\theta) = \max_{x \in G(\theta)} F(x, \theta)$ is the value function and $x^*$, $x^*(\theta) = \{x \in G(\theta) : F(x, \theta) = V(\theta)\}$ is the optimal policy correspondence.*

**Definition 4.** *Let $G : \Theta \to \mathbb{P}(\mathbb{R}^d)$ be a function from the parameter space $\Theta$ to the power set of $\mathbb{R}^d$. We say that $G$ is upper hemicontinuous at $\theta \in \Theta$ if $G(\theta)$ is nonempty and if, for every open set $U \subset \mathbb{R}^d$ with $G(\theta) \subset U$, there exists a $\delta > 0$ such that for every $\theta' \in N_\delta(\theta)$ (every $\theta'$ in some $\delta$-neighborhood of $\theta$), $G(\theta') \subset U$.*

In our set up $d = 2n$ and $\Theta = [0, \infty)$ such that $[\boldsymbol{f}\ \boldsymbol{g}] \in \mathbb{R}^{2n}$ and $\gamma \in \Theta$. Then $G(\theta)$ is the set of feasible $\boldsymbol{f}, \boldsymbol{g}$ and $V(\boldsymbol{f}, \boldsymbol{g}, \theta)$ is the DROT objective function.

**Proposition 2.** *Given two discrete measures $\mu, \nu$, a cost function $c$, and Bregman regularizers $\phi, \varphi$ and $\gamma^{-1} \in [0, \infty)$, the value function $V$ is well defined and continuous on $[0, \infty)$ and the optimal policy correspondence $x^*$ is also well defined and continuous on $(0, \infty)$. Furthermore, if $\phi, \varphi$ are both positive co-finite or both negative co-finite, then the optimal policy correspondence is upper hemicontinuous on $[0, \infty)$ and bounded from below.*

*Proof.* Let us start by defining a new problem $\mathrm{DROT}_n$ as follows. Here we add the following new constraints: $-n \leq \boldsymbol{f}_i, \boldsymbol{g}_j$. In this case, we have that the feasible region is bounded and closed and hence is compact.

We are going to show continuity using Berge's maximal theorem. Hence we need to show the assumptions for Berge's theorem are true. Here let $K_n = \{[\boldsymbol{f}, \boldsymbol{g}] \in \mathbb{R}^{2n} : -n \leq \boldsymbol{f}_i, \boldsymbol{g}_j\}$, then we have

$$X_n = \{[\boldsymbol{f}, \boldsymbol{g}] \in \mathbb{R}^{2n} : \boldsymbol{f}_i + \boldsymbol{g}_j \leq C_{ij}\} \cap K_n$$

This $X_n$ will be the feasible region for the problem $\mathrm{DROT}_n$. Now let $\Theta = [0, \infty)$. Now define $T : X_n \times \Theta \to \mathbb{R}$ that is defined as follows.

$$T(\boldsymbol{f}, \boldsymbol{g}, \gamma^{-1}) = \langle \boldsymbol{f}, \boldsymbol{a} \rangle + \langle \boldsymbol{g}, \boldsymbol{b} \rangle - \gamma^{-1} \phi(\boldsymbol{f}) - \gamma^{-1} \varphi(\boldsymbol{g})$$

Finally, let us define $G_n(\theta) = X_n$ for all $\theta \in \Theta$. In this case, we have that the value function is

$$V_n(\theta) = \max_{x \in G_n(\theta)} T(x, \theta),$$

where $x = [\boldsymbol{f}, \boldsymbol{g}]$ and the optimal policy correspondence is

$$x_n^*(\theta) = \{x \in G_n(\theta) : T(x, \theta) = V_n(\theta)\}.$$

The first few assumption for Berge's maximal theorem are that $T$ is a continuous function, $\Theta$ is closed and $X_n$ is closed. These are clearly true. Thus, we just need to show that $G$ is compact valued and continuous. First, we see that $X_n$ is compact. Hence $G$ is compact valued. Thus, we just need to show that $G$ is continuous.

We shall do this by showing that $G_n$ is upper and lower hemicontinuous.

For upper hemicontinuity, we need to show that for all $\theta \in \Theta$ that for every sequence $(\theta_j)_{j \in \mathbb{N}}$ with $\theta_j \to \theta$ and every sequence $(x_j)_{j \in \mathbb{N}}$ with $x_j \in G_n(\theta_j)$ for all $j$, there exists a convergent sub-sequence $x_{j_k}$ such that $x_{j_k} \to x \in G_n(\theta)$. In this case, since $G_n(\theta_j) = X_n$ for all $\theta_j$, we have that $x_j \in X_n$. Then since $X_n$ is compact, we have a convergent sub-sequence.

For lower hemicontinuity, we need to show that for all $\theta \in \Theta$, for every open set $X' \subset X_n$ with $G_n(\theta) \cap X' \neq \emptyset$, there exists a $\delta > 0$ such that for every $\theta' \in N_\delta(\theta)$, $G_n(\theta') \cap X' \neq \emptyset$. In this case, since $G_n(\theta) = X_n$ for all $\theta$, this is trivially true.

Thus, $G_n$ is compact valued and continuous. Thus, by the Berge's maximal theorem, we have that $V_n$ is well defined and continuous. Also we have that $x_n^*$ is upper hemicontinuous. Now we have that for a fixed $\theta \in (0, \infty)$, $[\boldsymbol{f}, \boldsymbol{g}] \to T(\boldsymbol{f}, \boldsymbol{g}, \theta)$ is a strictly concave function and $G_n(\theta)$ is a convex. Thus, we have that there has a unique maximizer. Thus, $x_n^*(\theta)$ is a singleton set. Thus, being upper hemicontinuous implies continuity and that the function $\theta \mapsto [\boldsymbol{f}, \boldsymbol{g}] \in x_n^*(\theta)$ is a continuous function.

Let $V, x^*$ be the value function and optimal policy correspondence for DROT. Then we need to show that $V, x^*$ are continuous at all $\theta \in (0, \infty)$. To do this let $\theta \in (0, \infty)$ and let $\boldsymbol{f}_{\phi,\varphi}^*, \boldsymbol{g}_{\phi,\varphi}^*$ be the optimal solutions. Then we know there exists an $n$ such that $[\boldsymbol{f}_{\phi,\varphi}^*, \boldsymbol{g}_{\phi,\varphi}^*] \in int(K_n)$. Thus, due to the continuity of $x_n^*$ there is a ball $B$ around $\theta$, such that $x_n^*(B) \subset int(K_n)$ and $x_n^* = x^*$ on $B$. Thus, $V = V_n$ on $B$. Thus, $V, x^*$ is continuous on $(0, \infty)$. Finally part 1 of Proposition 1 shows that $V$ is continuous at 0.

The final detail that we need to prove is the fact that $x^*$ is upper hemicontinuous at 0. First, suppose both $\phi$ and $\varphi$ are negative co-finite. Then since $\phi, \psi$ are bounded from below and

$$\phi(\boldsymbol{f}_{\phi,\varphi}^*) + \varphi(\boldsymbol{g}_{\phi,\varphi}^*) \le \phi(\boldsymbol{f}^*) + \varphi(\boldsymbol{g}^*).$$

We see that $\phi(\boldsymbol{f}_{\phi,\varphi}^*), \varphi(\boldsymbol{g}_{\phi,\varphi}^*)$ are bounded from above. Thus, since the two functions are negative co-finite, there exists an $N$ such that, $N \le \boldsymbol{f}, \boldsymbol{g}$. Thus, we see that, $x^* = x_N^*$. Thus, we have upper hemi-continuous at 0.

Let us now suppose that both $\phi$ and $\varphi$ are positive co-finite. Then since $\phi, \psi$ are bounded from below. and

$$\phi(\boldsymbol{f}_{\phi,\varphi}^*) + \varphi(\boldsymbol{g}_{\phi,\varphi}^*) \le \phi(\boldsymbol{f}^*) + \varphi(\boldsymbol{g}^*).$$

We see that $\phi(\boldsymbol{f}_{\phi,\varphi}^*), \varphi(\boldsymbol{g}_{\phi,\varphi}^*)$ are bounded from above. Thus, since the two functions are positive co-finite, there exists an $N$ such that, $N \ge \boldsymbol{f}, \boldsymbol{g}$.

Now we know that at $\gamma^{-1} \to 0$, we have that

$$\| \langle \boldsymbol{f}^* - \boldsymbol{f}_{\phi,\varphi}^*, \boldsymbol{a} \rangle + \langle \boldsymbol{g}^* - \boldsymbol{g}_{\phi,\varphi}^*, \boldsymbol{b} \rangle \| \to 0.$$

Thus now assume for the sake of contradiction that

$$\boldsymbol{f}_{\phi,\varphi}^* \to -\infty$$

as $\gamma^{-1} \to 0$. Then we have that

$$\langle \boldsymbol{f}^* - \boldsymbol{f}_{\phi,\varphi}^*, \boldsymbol{a} \rangle \to \infty$$

as $\gamma^{-1} \to 0$. Thus, we must have that

$$\langle \boldsymbol{g}^* - \boldsymbol{g}_{\phi,\varphi}^*, \boldsymbol{b} \rangle \to -\infty$$

as $\gamma^{-1} \to 0$. But then this would imply that there exists a coordinate of $\boldsymbol{g}_{\phi,\varphi}^*$ that goes to infinity as $\gamma^{-1} \to 0$. This is a contradiction. Thus $\boldsymbol{f}_{\phi,\varphi}^*$ is bounded from below.

Similarly, we have that $\boldsymbol{g}_{\phi,\varphi}^*$ is bounded from below. Thus, there exists an $N$ such that $x^* = x_N^*$. Thus, $x^*$ is upper hemicontinuous at 0. $\square$

The implication of the upper-hemicontinuity of the optimal policy correspondence is that any sequence of solutions $(\boldsymbol{f}_{\phi,\varphi}^*)_n, (\boldsymbol{g}_{\phi,\varphi}^*)_n$ to the DROT Problem 4 for a sequence of $(\gamma)_n$, has a convergent sub-sequence. Lower-hemicontinuity implies that all solutions to the OT Problem 2 can be expressed as limits of sequences of solutions to DROT.

Finally, we show that the transport map $\boldsymbol{P}$ that results from solving DROT is at least as sparse as that from the OT solution. While this is result is for the case when $\gamma$ is large, as we will see experimentally, we produce sparse solutions for all $\gamma$.

**Corollary 2.** *Suppose that we have an instance of Problem 2 such that for any two optimal dual solutions $(\boldsymbol{f}_1^*, \boldsymbol{g}_1^*), (\boldsymbol{f}_2^*, \boldsymbol{g}_2^*)$, we have that $\boldsymbol{f}_1^* - \boldsymbol{f}_2^* = c\boldsymbol{1}$, and $\boldsymbol{g}_1^* - \boldsymbol{g}_2^* = -c\boldsymbol{1}$. Then there exists $\Gamma$ such that for all $\gamma \ge \Gamma$, if $\boldsymbol{P}_\gamma^*$ is the solution to DROT Problem 4 for $\gamma$ and $\boldsymbol{P}^*$ is any optimal solution to Problem 2, then we have that $supp(\boldsymbol{P}_\gamma^*) \subset supp(\boldsymbol{P}^*)$.*

*Proof.* First, we note that the solution to the optimal transport problem is now unique upto constants. Then due to the existence of strictly complementary solutions. We see that all solution must be strictly complementary.

Let $\boldsymbol{f}^*, \boldsymbol{g}^*$ be the optimal solutions to the non-regularized problem. Then we know that $\text{supp}(\boldsymbol{P}^*) = \{i, j : \boldsymbol{f}_i^* + \boldsymbol{g}_j^* = \boldsymbol{C}_{i,j}\}$. Then there is an $\epsilon > 0$, such that for any non active constraint we have that

$$\boldsymbol{f}_i^* + \boldsymbol{g}_j^* - \boldsymbol{C}_{ij} < -\epsilon$$

Then let $V = \{\boldsymbol{f}, \boldsymbol{g} : \|\boldsymbol{f}^* - \boldsymbol{f}\| < \epsilon/3, \|\boldsymbol{g} - \boldsymbol{g}^*\| < \epsilon/3\}$. Then by upper continuity we know that there exists a $\delta > 0$ such that for all $\gamma^{-1} < \delta$ we have that $\boldsymbol{f}_{\phi,\varphi}^*, \boldsymbol{g}_{\phi,\varphi}^* \in V$. Thus, by complementary slackness we have the needed result.

$\square$

### 3.2 Example regularizers

In this subsection, we focus on three different example regularizers: quadratic, entropic, and exponential. All of these regularizers satisfy the theoretical assumptions of the theoretical analysis in the previous subsection although there are some important differences amongst them.

#### 3.2.1 Quadratic

The quadratic regularizers are $\phi(\boldsymbol{f}) = \|\boldsymbol{f}\|_2^2$ and similarly for $\varphi(\boldsymbol{g})$. This regularizer is thoroughly studied in Blondel et al. (2018) and, for brevity, we do not discuss it further. We observe that the regularizer is a co-finite Bregman function.

#### 3.2.2 Exponential

Let $\phi(\boldsymbol{f}) = \sum_{i=1}^n e^{\boldsymbol{f}_i}$ and similarly for $\varphi(\boldsymbol{g})$. We observe that $\phi, \varphi$ are positively co-finite Bregman functions and, by Theorem 1, this formulation of DROT must destroy mass. To be more concrete, the convex conjugate of $\phi$ is $\phi^*(\boldsymbol{x}) = \sum_{i=1}^n x_i \log(x_i) - x_i$ with the stipulation that $x_i \geq 0$ and similarly for $\varphi^*$. In Theorem 1, the variable $\boldsymbol{x}$ in the dual formulation of DROT is $\boldsymbol{x} = \boldsymbol{a} - \boldsymbol{P}\boldsymbol{1}_m$ and the requirement that $x_i \geq 0$ implies

$$\boldsymbol{a}_i - (\boldsymbol{P}\boldsymbol{1}_n)_i \geq 0 \quad \text{or} \quad \boldsymbol{a}_i \geq (\boldsymbol{P}\boldsymbol{1}_n)_i.$$

Hence, the transport process only destroys or preserves mass; it does not create it.

#### 3.2.3 Entropy

Let $\phi(\boldsymbol{f}) = \sum_{i=1}^n \boldsymbol{f}_i \log(\boldsymbol{f}_i) - \boldsymbol{f}_i$ and similarly for $\varphi(\boldsymbol{g})$. The convex conjugate of $\phi$ is $\phi^*(x) = \sum_{i=1}^n e^{x_i}$ and similarly for $\varphi^*$. In Remark 1, we detail the additional stipulations we impose when we use the entropic regularizers. These constraints include that $(\boldsymbol{f})_i, (\boldsymbol{g})_i \geq 0$ which implies that in the dual formulation of DROT, the variables $\boldsymbol{x}$ and $\boldsymbol{y}$ satisfy $\boldsymbol{x} = \boldsymbol{a} - (\boldsymbol{P}\boldsymbol{1}_m) - \boldsymbol{c}_1$ and $\boldsymbol{y} = \boldsymbol{b} - (\boldsymbol{P}^T\boldsymbol{1}_n) - \boldsymbol{c}_2$, where $\boldsymbol{c}_1, \boldsymbol{c}_2$ are vectors which non-negative entries. In the optimization problem, we optimize for $\boldsymbol{c}_1, \boldsymbol{c}_2$ as well. We minimize this term in the objective when $\boldsymbol{a} - (\boldsymbol{P}\boldsymbol{1}_n) - \boldsymbol{c}_1$ is negative, or when $\boldsymbol{a} < (\boldsymbol{P}\boldsymbol{1}_n) + \boldsymbol{c}_1$ (and similarly for $\boldsymbol{b}$). Because $\boldsymbol{c}_1$ is variable, it is not clear whether we favor creating or destroying mass. As we will see, however, in the experiments, we always favor creating mass in this formulation. This matches our intuition as $\boldsymbol{f}, \boldsymbol{g}$ must be positive.

### 3.3 Efficient algorithm: Project and Forget

While there are many different potential algorithmic techniques that could be used to solve this problem, we adopt a new algorithmic method, PROJECT AND FORGET Gilbert & Sonthalia (2020), which is a conversion of Bregman's cyclic method into an active set method and, as such, can solve highly constrained convex optimization problems. In Particular, Gilbert & Sonthalia (2020) performed an extensive comparison experiment to validate the choice of PROJECT AND FORGET for the quadratically regularized version of the

problem. They showed that using PROJECT AND FORGET, they could solve the problem for much larger values of $n$ for the quadratically regularized version of the problem.

PROJECT AND FORGET is an iterative method with three major steps per iteration: (i) an (efficient) oracle to find violated constraints, (ii) Bregman projection onto the hyperplanes defined by each of the active constraints, and (iii) the forgetting of constraints that no longer require attention. To better understand the algorithm the following discussion is adapted from Gilbert & Sonthalia (2020)

Let $L^{(\nu)}$ be the set of constraints in consideration at the start of the $\nu$th iteration. First, we run an oracle to add new constraints to get $\tilde{L}^{(\nu+1)}$. Then we do the Project and Forget steps.

The Project and Forget steps for the algorithm are presented in Algorithm 1. Let us step through the code to obtain an intuitive understanding of its behavior. Let $H_{ij} = \{\boldsymbol{f}, \boldsymbol{g} : \boldsymbol{f}_i + \boldsymbol{g}_j \leq \boldsymbol{C}_{ij}\} \in \tilde{L}^{(\nu+1)}$ be a constraint and $\boldsymbol{f}, \boldsymbol{g}$ the current iterate. The first step is to calculate $\boldsymbol{f}'\boldsymbol{g}'$ and $\theta$. Here $\boldsymbol{f}', \boldsymbol{g}'$ is the projection of $\boldsymbol{f}, \boldsymbol{g}$ onto the boundary of $H_{ij}$ and $\theta$ is a "measure" of how far $\boldsymbol{f}, \boldsymbol{g}$ is from $\boldsymbol{f}', \boldsymbol{g}'$. In general, $\theta$ can be any real number and so we examine two cases: $\theta$ positive or negative.

From Gilbert & Sonthalia (2020) we know that $\theta$ is negative if and only if the constraint is violated. In this case, in Algorithm 1 we have $c = \theta$ because the algorithm always maintains $z_{ij} \geq 0$. Then on line 5, we compute the projection of $\boldsymbol{f}, \boldsymbol{g}$ onto $H_{ij}$. Finally, since we corrected $\boldsymbol{f}\boldsymbol{g}$ for this constraint, we add $|\theta|$ to $z_{ij}$. Since each time we correct for $H_{ij}$, we add to $z_{ij}$, we see that $z_{ij}$ stores the total corrections made for $H_{ij}$. On the other hand, if $\theta$ is positive, this constraint is satisfied. In this case, if we also have that $z_{ij}$ is positive; i.e., we have corrected for $H_{ij}$ before and we have over compensated for this constraint. Thus, we must undo some of the corrections. If $c = z_{ij}$, then we undo all of the corrections and $z_i$ is set to 0. Otherwise, if $c = \theta$ we only undo part of the correction.

For the Forget step, given a constraint $H_{ij} \in \tilde{L}^{(\nu+1)}$, we check if $z_{ij}^{(\nu+1)} = 0$. If so, then we have not done any net corrections for this constraint and we can forget it; i.e., delete it from $\tilde{L}^{(\nu+1)}$.

If we think of $L^{(\nu)}$ as matrix, with each constraint being a row, we see that at each iteration $L^{(\nu)}$ is a sketch of the matrix of active constraints. Hence, during each iteration we update this sketch by adding new constraints (rows). During the Forget step, we determine which parts of our sketch are superfluous and we erase (forget) these parts (rows) of the sketch.

---

**Algorithm 1** Project and Forget algorithms.

---

1: **function** PROJECT$(x, z, L)$
2:     **for** $H_{ij} = \{\boldsymbol{f}, \boldsymbol{g} : \boldsymbol{f}_i + \boldsymbol{g}_j \leq \boldsymbol{C}_{ij}\} \in L$ **do**
3:         Find $\boldsymbol{f}', \boldsymbol{g}', \theta$ by solving $\theta\boldsymbol{e}_i := \nabla\phi(\boldsymbol{f}') - \nabla\phi(\boldsymbol{f})$, $\theta\boldsymbol{e}_j = \nabla\varphi(\boldsymbol{g}') - \nabla\varphi(\boldsymbol{g})$ and $\boldsymbol{f}', \boldsymbol{g}' \in H_{ij}$
4:         $c_{ij} = \min(z_{ij}, \theta)$
5:         $\boldsymbol{f}, \boldsymbol{g} \leftarrow \boldsymbol{f}_{new}, \boldsymbol{g}_{new}$, such that $\theta\boldsymbol{e}_i := \nabla\phi(\boldsymbol{f}_{new}) - \nabla\phi(\boldsymbol{f})$ and $\theta\boldsymbol{e}_j = \nabla\varphi(\boldsymbol{g}_{new}) - \nabla\varphi(\boldsymbol{g})$.
6:         $z_{ij} \leftarrow z_{ij} - c_{ij}$
    **return** $\boldsymbol{f}, \boldsymbol{g}, z$
7: **function** FORGET$(z, L)$
8:     **for** $H_{ij} = \{\boldsymbol{f}, \boldsymbol{g} : \boldsymbol{f}_i + \boldsymbol{g}_i \leq \boldsymbol{C}_{ij}\} \in L$ **do**
9:         **if** $z_{ij} == 0$ **then** Forget $H_{ij}$
    **return** $L$

---

To adapt PROJECT AND FORGET for DROT, the three major steps are as follows. First, we use a naive oracle that searches through all of the constraints and adds to the current list of active constraints any violated constraint. In particular, since each constraint is independently satisfied or not, we can do this search in parallel. In the project step, we observe that the constraints are of the form $\boldsymbol{f}_i + \boldsymbol{g}_j \leq \boldsymbol{C}_{ij}$. To calculate the projection, we first calculate $\boldsymbol{f}'_i, \boldsymbol{g}'_j, \theta$ as the solutions to the following equations, where $\boldsymbol{e}_i, \boldsymbol{e}_j$ are the $i, j$th standard basis vectors.

$$\theta\boldsymbol{e}_i := \nabla\phi(\boldsymbol{f}') - \nabla\phi(\boldsymbol{f}) \text{ and } \theta\boldsymbol{e}_j = \nabla\varphi(\boldsymbol{g}') - \nabla\varphi(\boldsymbol{g}).$$

We discuss an analytic formula for $\theta$, that only depends on $\boldsymbol{f}_i, \boldsymbol{g}_j, \boldsymbol{C}_{ij}$ for the different regularizers below.

**Quadratic regularizer.** For the quadratic regularizer, we have that $\theta = \frac{C_{ij} - f_i - g_j}{2\gamma}$.

**Entropic regularizer.** In the case of the entropic regularizer, we have that

$$\theta = \log\left(\frac{C_{ij}}{f_i + g_j}\right)/\gamma.$$

And for the exponential regularizer, $\theta$ is given by

$$\theta = \frac{-e^{f_i} - e^{g_j} + \sqrt{(e^{f_i} - e^{g_j})^2 + 4e^{C_{ij}}}}{2\gamma}.$$

For exponential, this is done by solving the Lagrange multiplier problem.

**Mixed regularizers.** In the case we want to mix regularizers, calculating $\theta$ is more difficult. For example, if $\phi$ is quadratic and $\varphi$ is entropy, then $\theta$ is the root of $e^x + x + f_i + g_j - C_{ij}$.

Once we have calculated $\theta$, we set $c := \min(P_{ij}, \theta)$ and we update $P_{ij} \leftarrow P_{ij} - c$ and $f, g$ as follows

$$f \leftarrow \nabla\phi^{-1}(ce_i + \nabla\phi(f)) \text{ and } g \leftarrow \nabla\varphi^{-1}(ce_j + \nabla\varphi(g)).$$

In the forget step, if $P_{ij} = 0$, then we forget the related constraint (i.e., remove it from the list of active constraints). Note that $P$ is the dual variable and is the desired transportation plan. One feature of PROJECT AND FORGET is in addition to calculating the primal variables, we also retain the desired dual variable $P$, the transportation plan.

One of the reasons we chose to solve DROT with PROJECT AND FORGET is for its convergence analysis and rate. Specifically, we see that each iteration of the method takes $O(n^2)$ time and the method uses a total of $O(n^2)$ memory. Further, Gilbert & Sonthalia (2020) show that PROJECT AND FORGET has a linear rate of convergence and that the rate is at most $\frac{L}{L+\mu^2}$ for some $\mu \in (0, 1]$, where $L$ is the number of active constraints. Corollary 2 gives us an estimate of the sparsity of our solutions $P^*_{\phi,\varphi}$ and, hence, an estimate on the number of active constraints. (We note that there are comparatively few active constraints typically). Thus, giving us a reasonable problem specific upper bound on the rate of convergence.

## 4 Experiments

In this section, we provide extensive experimental evidence to support the theoretical results presented in the previous section, to provide the intuition about dual regularized optimal transport (where theoretical analysis is unavailable), and to demonstrate that our new formulation of optimal transport is both different and useful (performing domain transfer tasks, including color transfer and digit classification).[2]

### 4.1 Verifying theoretical properties

The first solution property that we verify experimentally is the sparsity of the transport plan. To generate a problem instance for verification, we uniformly sample two distributions $a, b$ from $\Delta^{100}$. Then we sample $C_{ij}$ independently and uniformly from $[0, 1]$. As we can see from Figure 1(i), in all cases, we find solutions that are sparser than the true optimal transport plan. As the regularization parameter $\gamma$ increases, the size of the support of our transport plans increases until we reach the true support size. For the entropic and exponentially regularized versions, for $\gamma = 10^5$, the optimization had not converged so we do not plot those results.

Next, we evaluate how well our objective functions approximate the Wasserstein distance (the objective of Problem 1) and how well our transport plans approximate the true plans. We construct a simple problem instance (as our previous instance is difficult to calculate for large $\gamma$) consisting of two Gaussian distributions with means $\pm 15$ and variance 10. The cost matrix $C$ is given by $C_{ij} = 1$. Then we plot

---

[2]All experiments were run on a machine with 8 cores and 56 GB of memory.

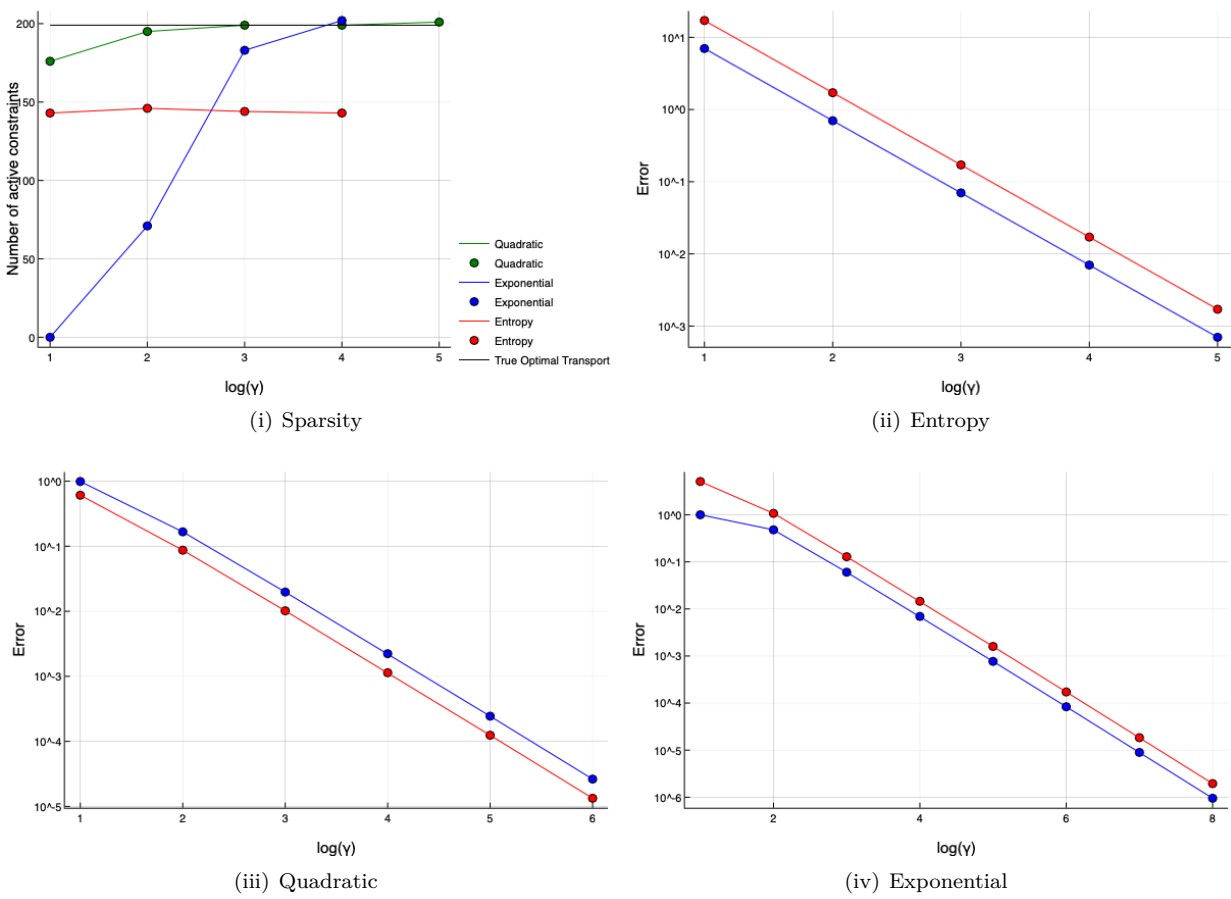

Figure 1: (i) Sparsity of the solutions for the different regularizers versus the regularization parameter; (ii–iv) Error $|\langle \boldsymbol{C}, \boldsymbol{P}^* - \boldsymbol{P}^*_{\phi,\varphi}\rangle|$ (blue line) and $OT(\boldsymbol{a}, \boldsymbol{b}) - DROT(\boldsymbol{a}, \boldsymbol{b})$ (red line) versus $\gamma$ for the three different regularizers.

$OT(\boldsymbol{a}, \boldsymbol{b}) - DROT(\boldsymbol{a}, \boldsymbol{b})$ (red line) and $|\langle \boldsymbol{C}, \boldsymbol{P}^* - \boldsymbol{P}^*_{\phi,\varphi}\rangle|$ (blue line) versus $\gamma$. Furthermore, the gap between the two lines is $\phi^*(\gamma * (\boldsymbol{a} - \boldsymbol{P}\boldsymbol{1}_m)/\gamma + \varphi^*(\gamma * (\boldsymbol{a} - \boldsymbol{P}^T\boldsymbol{1}_n)/\gamma$. From our theoretical analysis, we know that all of these quantities should be $O(\gamma^{-1})$. From the plots it is evident that $OT(\boldsymbol{a}, \boldsymbol{b}) - DROT(\boldsymbol{a}, \boldsymbol{b})$ (red line) and $|\langle \boldsymbol{C}, \boldsymbol{P}^* - \boldsymbol{P}^*_{\phi,\varphi}\rangle|$ (blue line) decrease linearly. Finally, since the plots are log-log plot, the plots show that $\phi^*(\gamma * (\boldsymbol{a} - \boldsymbol{P}\boldsymbol{1}_m))/\gamma + \varphi^*(\gamma * (\boldsymbol{a} - \boldsymbol{P}^T\boldsymbol{1}_n))/\gamma$ also decrease linearly with respect to $\gamma$. Thus, the experiments suggest that the theoretical error rate is tight. That is the error is $\Theta(\gamma^{-1})$.

Finally, we test the intuition sketched in our theoretical analysis as to when mass is created versus destroyed. Specifically that, entropy regularization creates mass, the exponential regularization destroys mass, and the quadratically regularized problem does both. To verify this, we uniformly sample two distributions $\boldsymbol{a}, \boldsymbol{b}$ from $\Delta^{100}$ and sample $\boldsymbol{C}_{ij}$ independently and uniformly from $[0, 1]$. Then we compute the transport plan and marginals for all three different regularizers for a variety of different values of $\gamma$. Figure 2 shows that our intuition matches exactly what occurs in practice. The quadratic regularizer both creates and destroys mass; that is, sometimes the yellow bars (bar chart for $\boldsymbol{a}$) are bigger and sometimes the yellow bars are smaller. The exponential regularizer only destroys mass; i.e., the yellow bars are always bigger. Finally, the entropic regularizer only creates mass; i.e., the yellow bars are always smaller. In each case, we see that as $\gamma$ gets bigger, the marginals of the transport plan better approximate the true marginals.

Note that all experiments were run until the PROJECT AND FORGET feasibility error was smaller than 1e-15.

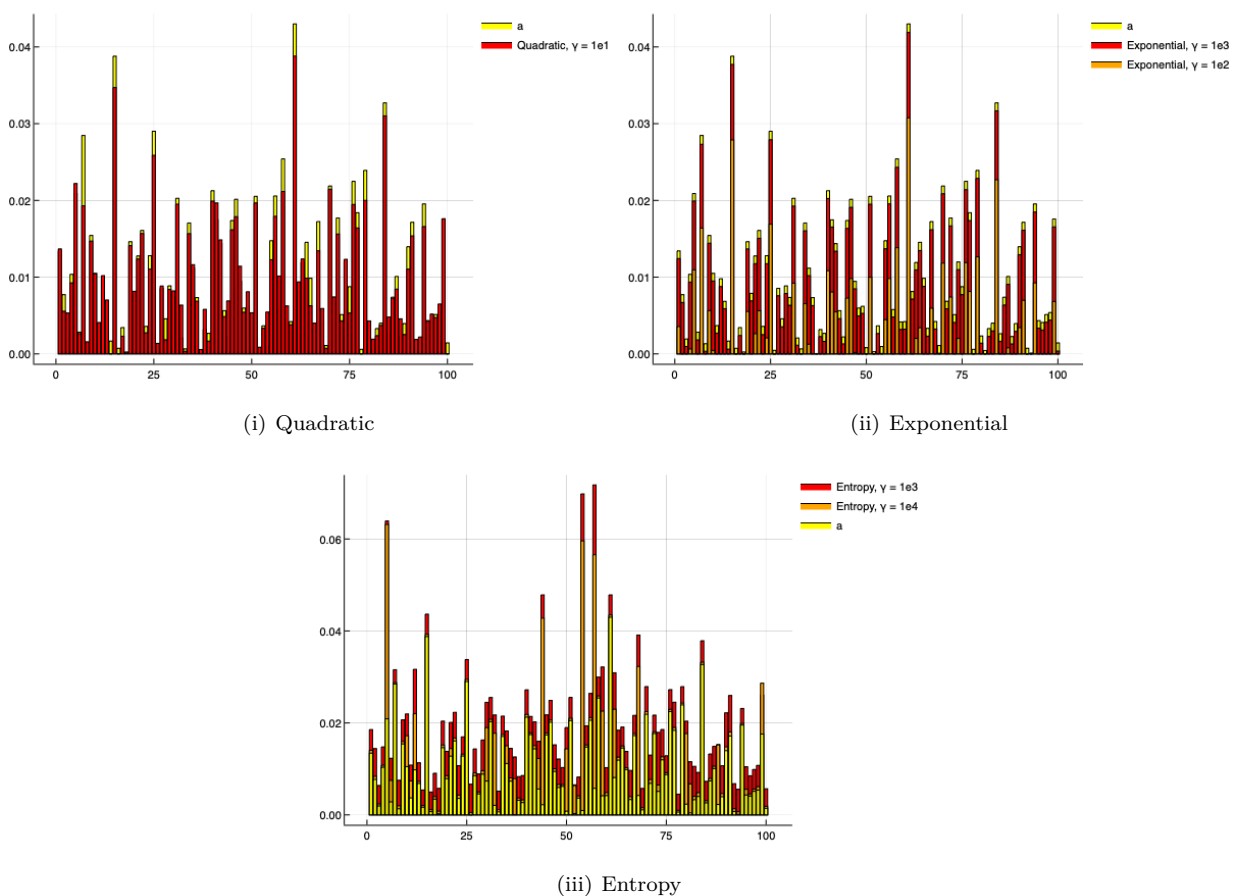

(i) Quadratic

(ii) Exponential

(iii) Entropy

Figure 2: Graphs showing the mass creation and destruction for the different regularizers. The yellow bars represent the true marginal distribution.

## 4.2 Domain Transfer

In this section, we explore how our new formulation performs on the task of domain transfer. We also investigate our intuition as to how the different regularizers affect the results. The goal of this section is not to present state of the art results for domain transfer, but to demonstrate that creating versus destroying mass gives us different results. And so, being able to decide whether mass is created or destroyed is a desirable attribute in a problem formulation and algorithmic method.

We compare our formulation DROT against other formulations. Specifically, we compare against standard optimal transport OT, entropic regularization of the primal ROT, and UOT with entropic regularization of the primal with KL divergence controlling the deviation from marginals. All of our DROT formulations are solved using PROJECT AND FORGET. The other formulations are solved using the python optimal transport library with the following algorithms: we solve ROT using the algorithm in Cuturi (2013), we solve OT using the algorithm in Bonneel et al. (2011), and we solve UOT using the algorithm in Chizat et al. (2016).

For all domain transfer problems we use the squared Euclidean distance as the cost function. Thus, once we have computed our transport plans $\boldsymbol{P}$ (obtained from solving any of the versions of optimal transport), we compute the barycentric projection map to transfer one data set into the domain of the other data set. That is, because we use the squared Euclidean distance as the cost, if $\boldsymbol{a}, \boldsymbol{b}$ are the two data sets, the transport of $\boldsymbol{a}$ to the domain of $\boldsymbol{b}$, denoted $\hat{\boldsymbol{a}}$ is given by, $\hat{\boldsymbol{a}}_i = \frac{\sum_{j=1}^{n} \boldsymbol{P}_{ij} \boldsymbol{b}_j}{\sum_{j=1}^{n} \boldsymbol{P}_{ij}}$.

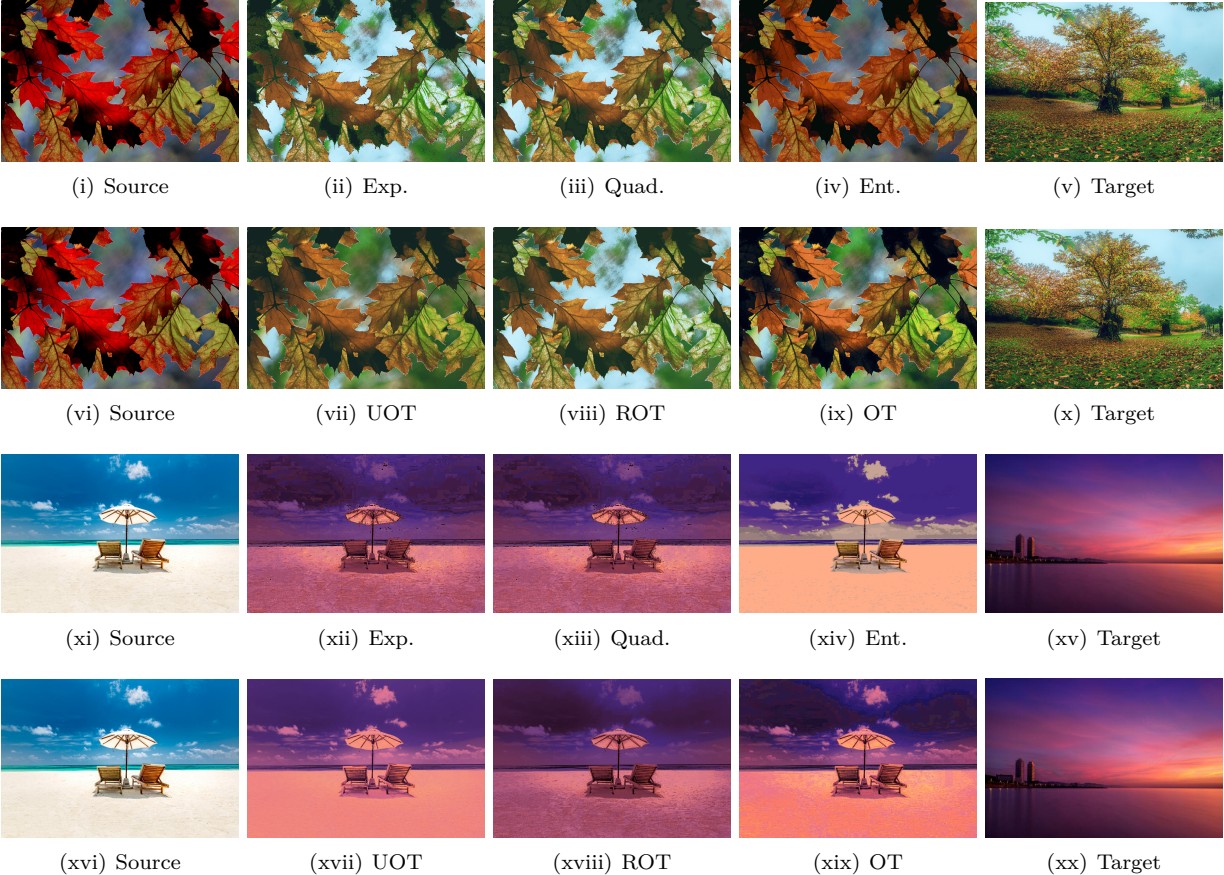

Figure 3: Images produced by doing color transfer using different regularizers (Exponential, Quadratic, Entropy) for DROT and images produced by doing color transfer using other formulations of optimal transport.

### 4.2.1  Color Transfer

Color transfer consists of the first domain transfer experiment. In these experiments, we use the same setup as Blondel et al. (2018). For each picture, we first perform $k$ means to cluster the three dimensional pixels in each image, generating $k$ color centers for each image. We used $k = 4096$ clusters. These centers are the point masses for the two distributions. The weight of each center is proportional to the number of points assigned to that cluster and the cost matrix is given by the Euclidean squared distance between the color centers. We want to demonstrate two things with this experiment:

1. DROT results in good quality images that look different. This is not the case with the other formulations of OT, which produce similar pictures, as seen in Figure 3.
2. The regularization parameter $\gamma$, when used with the quadratic regularizer, destroys mass and this is evident in the images but when used with the entropic regularizer, the way in which mass is created is not reflected in the images (although it is in a toy example).

For the quadratic regularizer $\gamma = 1e4$, for the entropic regularizer $\gamma = 1e4$, for the exponential regularizer, $\gamma = 10^{log_{10}(e^{10})} \approx 10^{4.34}$. Here we picked $\gamma$ that looked best for the first set of images and used the same $\gamma$ for the second set. For ROT we set $\gamma = 1e - 2$. For UOT we set the regularizer $\gamma_1 = 1e - 2$, and we set the penalty $\gamma_3 = \gamma_2 = 1e1$.

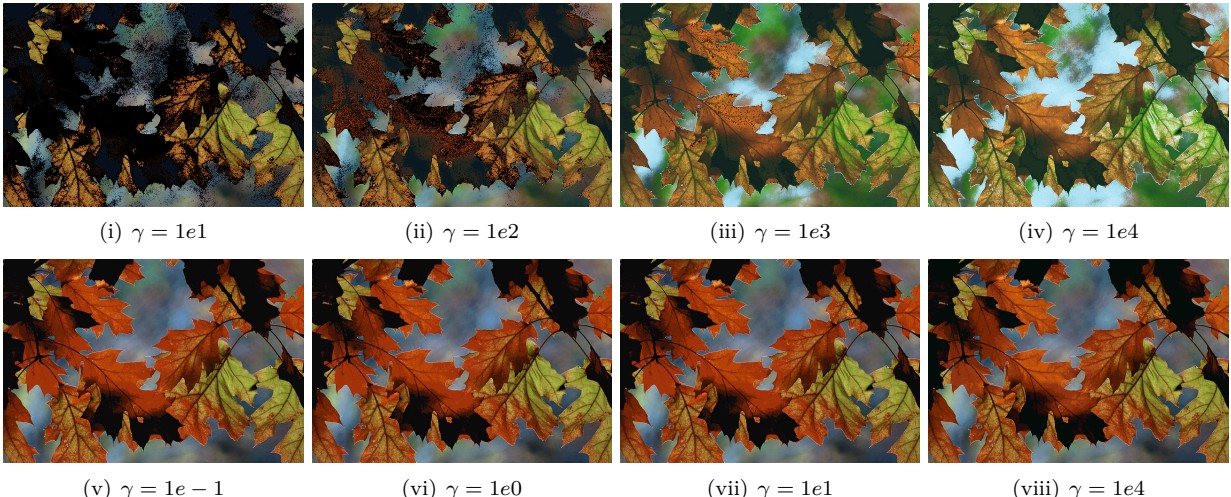

(i) $\gamma = 1e1$      (ii) $\gamma = 1e2$      (iii) $\gamma = 1e3$      (iv) $\gamma = 1e4$

(v) $\gamma = 1e-1$      (vi) $\gamma = 1e0$      (vii) $\gamma = 1e1$      (viii) $\gamma = 1e4$

Figure 4: Images produced by doing color transfer for different values of $\gamma$. The top row is for the quadratic regularizer, and the bottom row is for the entropic regularizer.

For the first demonstration, we can see the performance of the different regularizers in Figure 3. If we use the entropic regularizer, then the transferred image is more faithful to the original color distribution. Additionally, we see that entropic regularized images are cleaner and have fewer artifacts.

For the second demonstration, we can see from Figure 4, that when $\gamma$ is small and we use the quadratic regularizer, we tend to destroy the mass; i.e., the images are corrupted. As we can see in Figure 4, however, for the entropic regularizer, for all values of $\gamma$, the images look identical. We argue that this phenomenon occurs as a result of two different phenomena. First, we note that the entries in the cost matrix are less than 1. Because of the entropic regularizer, the critical point of the objective function always has the entries greater than 1. Thus, the solution to the entropic regularized problem will always be on the boundary, regardless of the value of $\gamma$. That is, mass transport always occurs. This does not, however, explain why the images look identical. We conjecture a second phenomenon is at play: when we have a convex cost function, we conjecture that, changing $\gamma$ results in creating mass simply by shifting the distribution upwards (as demonstrated in Figure 5). That is, the transport plan maintains the shape of the distribution and just shifts it up. For images, shifting the distribution by a bounded amount does not impact the appearance of the color transfer and the images look similar.

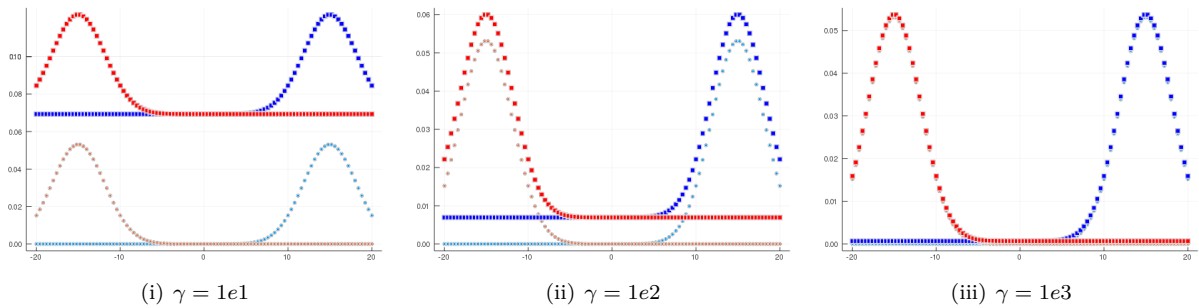

(i) $\gamma = 1e1$      (ii) $\gamma = 1e2$      (iii) $\gamma = 1e3$

Figure 5: Graphs showing that the entropic regularizer maintains the distribution shape. We used the squared Euclidean distance as the cost function and performed transport from the red distribution to the blue distribution.

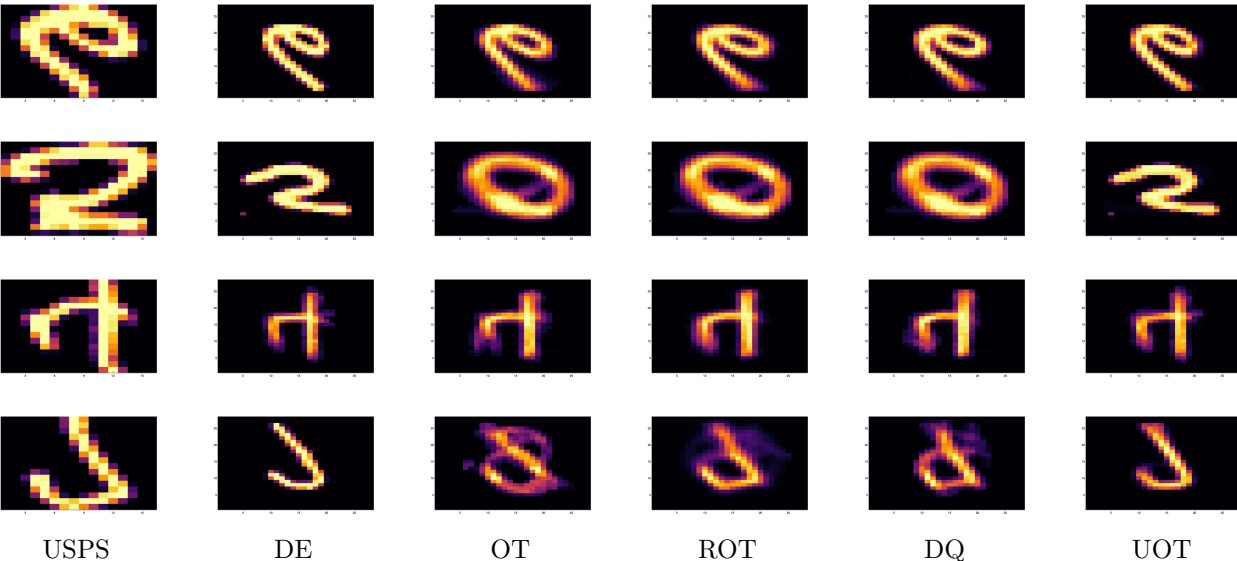

Figure 6: Images of the first four digits in the USPS dataset, when transported using to the MNIST domain using various optimal transport problems. DE/DQ refers to entropic/quadratic regularized version of DROT.

| Problem | Trained on MNIST | Trained on USPS |
|---|---|---|
| Dual Entropy | 76.46% | 62.54% |
| Dual Quadratic | 65.75% | 63.79% |
| OT | 62.04% | 65.32% |
| UOT | 75.44% | 66.16% |
| ROT | 66.99% | 63.87% |

Table 1: Accuracy using a 1 nearest neighbor classifier after transporting the USPS dataset to the MNIST domain.

### 4.2.2 MNIST, USPS classification

Finally, we use domain adaptation for classification. To test the performance of DROT, we transport between the MNIST training data set and USPS training data sets. First, we pad the USPS images with zeros so that they are are the same size as the MNIST images and the use the squared Euclidean distance as the metric between the two data sets. We then transport the USPS training set images to the MNIST domain.

For the quadratic regularizer, we set $\gamma = 1e7$, the entropic regularizer we set $\gamma = 1e5$. These were the smallest $\gamma$'s at which transport happened. For ROT and UOT we set $\gamma = \gamma_1 = \gamma_2 = \gamma_3 = 1$. Note $\gamma$ was finalized before we looked at any of the digits or the prediction accuracy. It was chosen whenever the transport plan $\boldsymbol{P}$ had non trivial number of non-zero entries.

Let us start by examining the appearance of the transported digits. Figure 6 shows what the first 4 digits in the USPS data set look like after they have been transported to the MNIST domain. We can see again that the entropic regularized transport is the most faithful to the original image and has the cleanest new digits. We note that some of the digits are flipped, we present them this way as they are flipped in the dataset we have as well. We then use the various transported USPS digits for classification. We use labels for both MNIST and USPS. That is, we try to classify the MNIST digits using a classifier trained on the transported USPS dataset and to classify the transported USPS digits using a classifier trained on the MNIST dataset. Table 1 shows that the entropic regularized version performs well.

## 5 Conclusion

In conclusion, in this paper, we present a new formulation of optimal transport called Dual Regularized Optimal Transport. We prove many theoretical results, including connections to the UOT problem, and properties of the solutions for various choices of $\phi, \varphi, \gamma$. We also build the intuition that regularizing $\boldsymbol{f}, \boldsymbol{g}$ to be more positive results in mass creation and regularizing to be more negative results in mass destruction. We support this intuition with experimental evidence. Finally, we also showed creating mass, via the dual entropic regularized problem results in novel, and useful results when applied to domain transfer.

We hope that this paper helps understand the dual problem of optimal transport and that future researchers can use the useful interpretations of the dual to design optimal transport based algorithms for their various different applications. In terms of future work, we are interested in understanding under what situations such problems result in a metric on the space of distributions. We are also interested in dynamical formulations of the problem as well.

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
