# OpenReview forum: "Dual Regularized Optimal Transport"
_TMLR — Rejected by TMLR_

### Review · Reviewer_QDap · 2022-09-05

**Summary Of Contributions:**

In this work, the authors introduce a regularized (dual) OT problem in finite dimension (i.e. discrete measures) which takes the following équivalent primal-dual forms:

$$\max_{f,g : f \oplus g \leq C} f^T a + g^T b - \epsilon (\phi(f) + \varphi(g)) = \min_{P \geq 0} C^T P + \epsilon \left( \phi^*( (a - P 1)/\epsilon) + \varphi^*((b - P^T 1) / \epsilon) \right), $$

where $C \in \mathbb{R}^{n \times m}$ is a cost matrix, $\phi, \varphi$ are chosen regularizers (e.g. quadratic, entropic, etc.) and $\epsilon = \frac{1}{\gamma}$ is the regularization parameter, so that in the regime $\epsilon \to 0$ ($\Leftrightarrow \gamma \to \infty$) we expect to retrieve the usual (unregularized) OT problem. Here, $A^*(p) = \sup_x p^T x - A(x)$ denotes the Legendre conjugate of a convex function $A$.

Aside from the introduction of the problem (and the primal-dual equivalence), the authors support their work with the followings :
- [Proposition 1] A control in $\epsilon$ of the "approximation error" in terms of both objective value and optimizers, along with a convergence result in the regime $\epsilon \to 0$ [Prop 2].
- [Corollary 2] A "sparsity" result (in sense that when $\epsilon = 0$, the usual optimal transport plan $P^*$ is known to be fairly sparse) that reads, for $\epsilon$ small enough, $\mathrm{support}(P^*_\epsilon) \subset \mathrm{support}(P^*)$.
- The introduction of a new algorithm in the context of computational optimal transport (to the best of my knowledge) : Project and Forget.
- An extensive set of numerical experiment to showcase the practical use of the proposed approach.

**Broader Impact Concerns:**

I did not identify any Ethical concern _specific_ to this work.

**Requested Changes:**

I encourage the authors to revise their work taking into account the previous remarks, in particular the section 3 regarding the rigor of mathematical statements done through the paper, which is clearly critical.

Section 1 and 2 above are also quite important in my opinion.

Naturally, fixing typos would help to make the flow of the paper smoother.

**Strengths And Weaknesses:**

# Strengths:

- Nicely illustrated experiments.
- Interesting interpretation of the role played by the regularization function $\phi, \varphi$ in terms of mass creation/destruction.
- Refreshing introduction of a new algorithm to address computational OT problems.

# Weaknesses:

## 1. Novelty with respect to existing unbalanced OT models.

While I acknowledge that acceptance at TMLR should be based on objective paper soundness rather than subjective significance criteria, I feel that the model introduced in this work lacks true novelty. While the works mentions the seminal papers of Chizat et al. / Liero et al. (2015) when it comes to Unbalanced OT, I think that more recent works such at the one of Séjourné et al., _Sinkhorn divergences for unbalanced OT_ (2019) should be discussed as well. In this work, Séjourné and co-authors introduce a primal Unbalanced Regularized OT problem model of the form
$$ \min_\pi \braket{\pi,c} + \epsilon \left(D_\phi(\pi_1 | \alpha) + D_\varphi (\pi_2 | \beta) \right) + \epsilon' \mathrm{KL}(\pi | \alpha \otimes \beta),$$
where $D_\phi(\mu | \nu) = \int \phi(d \mu / d \nu) d \nu$ is a general Czisar divergence (and $\pi_1,\pi_2$ are the marginals of the transport plan $\pi$). In particular, in discrete settings, and with $\epsilon' = 0$, it seems that one essentially retrieves the setting of this work. For instance, with $\phi(z) = |z - 1|^2$, I think that $D_\phi( \cdot )$ boils down to the quadratic cost, etc. This work also puts a significant effort in theoretical considerations (in particular, it encompasses continuous settings which are of theoretical interest). While I agree that the present paper has different aims (it is much "computational OT oriented"), the difference between these work should be discussed in depth.

## 2. Discussion of related works

In the same vein, I think some other works should be discussed in a dedicated Related Work section, just to make more salient the contributions of the current work. In particular, the work _Semidual Regularized OT_ of Peyré and Cuturi should be discussed, just to make clear that the proximity in the titles is actually misleading: the work of Peyré and Cuturi is considering adding an entropic regularization to the dual (and only consider probability measures) while the present paper is adopting the "marginal regularization" approach. (Note that, to me, Séjourné is unifying the two approaches.)

## 3. Mathematical rigor can be improved

Concurrent works/significance put aside, I think that the paper has room for improvement in terms of mathematical (and writing) rigor. There are different aspects of claims/proofs that I struggled to proofread. In particular:
- References to claim/results/models of other paper should be more precise. For instance, in the claim "this is in contrast with the formulation of Liero et al.", an Eq°/page number would have been appreciated, given that the paper is more than 100 pages long. Same holds for the use of results of Rockafellar (1970) manuscript (the section should be provided at least), the mention of the "Berge's maximal theorem" without reference, etc.
- Some statements lack rigor. For instance, in Definition 1 (co-finiteness), the notation $x \geq 0$ is somewhat ambiguous for $x \in \mathbb{R}^n$ (I guess this is component-wise). The claim $f(rx) / r \to \infty$ when $r \to \infty$ is also unlikely to hold when $x = 0$, so I guess it is $x \geq 0 \text{ component-wise, and } x \neq 0$. Perhaps would it be simpler to introduce the (more standard, I'd say) definition of super-coerciveness (see https://carmamaths.org/resources/jon/Preprints/Books/CUP/cup-final.pdf Prop 3.5.4), reading $f(x)/\|x\| \to \infty$ when $\|x\| \to \infty$.
- I failed to check some proofs/claims, which I partially attribute to some inaccuracies/missing arguments. For instance:
    - In the proof of Corollary 1, it is claimed that "due to the convexity of $\phi$, (...) $\phi(f^*_{\phi,\varphi})$ is bounded from above". It is not said _in which variable_ this quantity is bounded. I guess it should be $\gamma$, but part 1 of Prop 1 only tells us that $\gamma \mapsto \phi(f^*_{\gamma}) + \varphi(g^*_{\gamma})$ is bounded (I let $f^*_{\phi,\varphi} = f^*_{\gamma}$) to stress the dependence on $\gamma$). One must explain why $\varphi$ is lower bounded (which would in turn give that $\gamma \mapsto \phi(f^*_{\gamma})$ is upper bounded). I guess this has to do with co-finiteness but this has to be explained. Similarly, in the proof of Prop 2, it is said that "(...) this would imply that there exists a coordinate of $g^*_{\gamma}$ that goes to $\infty$ as $\gamma^{-1} \to 0$. This is a contradiction." Why?
    - The paper claims in some places that "since convex functions are bounded from below...", which is obviously wrong (take $t \mapsto \frac{1}{t} - t$ on $\mathbb{R}_+^*$). I guess authors implicitly use the co-finiteness assumption here; please make it explicit.
    - In the proof of corollary 2, it is claimed that "we know that $\mathrm{support}(P^*) = \{i,j:\ f^*_i + g^*_j < C_{i,j}\}$. Can you give a reference for this claim? According to Eq (2.23) and Prop 3.2 in _Computational Optimal Transport_ (Peyré & Cuturi), the opposite is true actually: the support of $P^*$ belong to the set $f \oplus g = C$.
- I do not understand the presentation _Project and Forget_ algorithm. In particular,
    - In which space does the parameter $\theta$ live? Given the following equations linking $\theta$ to $f_i,g_j$, I guess it belongs to $\mathbb{R}^{n \times m}$. Am I correct? If so, please write $\theta_{i,j}$ when appropriate. Is $\theta$ simply a scalair? (In which case I do not understand how the following analytic formula could hold).
    - What means the relation $\theta e_i = \nabla \phi(f') - \nabla \phi(f)$ ? Does $\theta e_i$ represent the $i$-th column of $\theta$ (seen as a matrix-vector product)? If so, shouldn't the second equation (involving $\varphi(g)$) be $\theta^T e_j$?
    - From where does the "analytic formula for $\theta$" come?

# Other minor remarks/questions/suggestions:

- Mathematical terminology tends to be inconsistent through the paper. For instance, $\phi^*$ is first referred to as the _convex conjugate_ of $\phi$, and becomes the _convex dual_ in some places. In corollary one, the notation $P^*_{\gamma}$ is used, but it becomes $P^*_{\phi,\varphi}$ in the proof (in some places).
- If possible, the citation style should be of the form (Author, year) when citing a paper instead of Author (year). This would avoid weird things like "Cuturi Cuturi (2013)...".
- While I think that I mostly understand the proof of Proposition 2, it extensively use the fact that $G_n(\theta_j)$ does not depend on $\theta_j$, which simplifies things a lot. Isn't there a more adapted statement to get the result in that way-simpler setting? Here the hammer seems a bit bigger for the nail.
- A missing-right-parenthesis typo occurs in many places.
- It may be handy to provide the definition of _convex conjugate_ for the reader unfamiliar with convex analysis.
- What is meant by "the solution to Problem 2 (...) is sparse (...) which means that (...) the solutions are interpretable"?
- In 2.1, I do not exactly understand the statement "(...) it is unclear how the solution methods balance creation, destruction, and transportation of mass". Isn't it the role of the parameter $\gamma_1,\gamma_2,\gamma_3$? For instance, if $\gamma_2 = \gamma_3 = 0$, you allow for free mass destruction/creation, so basically you won't transport anything (assuming $C \geq 0$). Conversly, in the regime $\gamma_2, \gamma_3 \to \infty$, you pay a lot for violating the marginal constraints so that you retrieve the exact (regularized) OT problem.
- It could have been handy to introduce the standard Dual OT problem before section 2.2 to make DROT "more natural" to the unfamiliar reader.
- The economic interpretation proposed in the paper (attributed to Cuturi and Peyré, 2018) can actually be found in (Villani, 08). Furthermore, the relation $f_i + g_j \leq C_{i,j}$ is not simply "some cost contrain" in economic terms. It means that, as a shipper, we cannot propose to handle supply at price $f_i$ at $i$ and distribute them at price $g_j$ at $j$ at a cost higher than the transport cost $C_{i,j}$ for the producer (the producer won't call for our shipping service if it is more expensive than what they can do by themselves).
- In the proof of Theorem 1, can't we just use in the Lagrangian (once min and max have been inverted) that $\min_f \braket{f, a - P 1} - \frac{1}{\gamma} \phi(f) = \phi^*(\gamma (a - P1)) / \gamma $ by definition of the convex conjugate?
- Since the parametrization is more about $\gamma^{-1}$ than $\gamma$ (e.g. in Prop 2), wouldn't it simplify things to directly work with $\epsilon = \gamma^{-1}$?
- In the proof of Prop 2, there is a typo of sign in the definition of $T$.
- In the proof of Corollary 2, shouldn't it be "Let $f^*, g^*$ be the optimal solutions to the **non-** regularized problem"?
- In the experimental section, Figure 1, (i), it seems that for large $\gamma$, $P^*_{\gamma}$ has a larger support than $P^*$. It contradicts Corollary 2; is it due to numerical errors? Furthermore, this figure actually suggest that the smaller the $\gamma$, the sparser the transportation plan; do you have any intuition if this claim is correct and, if so, how to prove it?

---

> ### Author Response · Authors · 2022-09-27
> **Response**
>
> **Questions About Project and Forget**
> ---
>
> Here $\theta \in \mathbb{R}$ and is a scalar that depends on the current constraint and iterate. Here $e_i$ is the $i$th standard basis vector (i.e., has a 1 in the $I$th row and 0s otherwise).
>
> In terms of the analytic formula for $\theta$ given specific functions, $\phi, \psi$, this gives us a system of equations which in some cases can be solved analytically. For example, in the quadratic case the gradient is linear. So we have linear system of equations which can be solved.
>
> **Proof Claims**
> ---
>
> 1) The reviewer is correct that the convex functions don't have to be bounded from below. In fact, I think for this proof, the positive co-finiteness assumption is not enough either. We will need to add an assumption to the results.
>
> Here we get a contradiction, because we note that $\psi(g_{\phi,\psi}^*)$ is upper bounded, then if a coordinate of $g_{\phi,psi}^*$ goes to infinity, due to the function being coercive we get a contradiction.
>
> 2) The reviewer is correct. We are implicitly making use of the assumption that the functions go to infinity. We have made this change in the modified version.
>
> 3) Support of $P^*$. The reviewer is correct and that is a typo on our part. However, the result is still valid, as we what we actually want to say is that our solutions have 0s whenever $P^*$ has a zero. So, we should be looking at the complement of the support of $P^*$.
>
> **Other Rigor Issues**
> ---
>
> Thank you for pointing out the missing references. These have now been added.
>
> At the time of writing, we had not seen the term coercive used but had seen co-finiteness used in prior work and hence used that terminology.
>
> **Related Works**
> --
>
> Thank you for pointing us to Séjourné et al. paper. We had not seen this before. We would need to read this carefully and will then add in the relevant comparison.
>
> **Minor Comments**
> ---
>
> Thank you for the feedback we shall make these changes.

---

> ### Author Response · Authors · 2022-09-28
> **Response to Questions in Minor Comments**
>
> **"While I think that I mostly understand the proof of Proposition 2,...Here the hammer seems a bit bigger for the nail."**
>
> While it might be possible to avoid this, we currently do not have a proof.
>
> **What is meant by "the solution to Problem 2 (...) is sparse (...) which means that (...) the solutions are interpretable"?**
>
> Here we mean that when solutions are sparse they are similar to matchings. Which seems to be more interpretable. See https://aclanthology.org/2020.acl-main.496.pdf for an example.
>
> **In 2.1, I do not exactly understand the statement "(...) it is unclear how the solution methods balance creation, destruction, and transportation of mass". ... you pay a lot for violating the marginal constraints so that you retrieve the exact (regularized) OT problem."**
>
> Yes that is true, in the sense that this lets us control the magnitude of the deviation. However, It does not let us control the sign of the deviation from the marginal.
>
> **"In the experimental section, Figure 1, (i), it seems that for large"**
>
> That is a numerical issue unfortunately. The reviewer is right in that the experiment actually shows more than the statement proved. However, we do not have a proof for the stronger statement and hence have only proved the weaker statement in the paper.

---

> > ### Comment · Reviewer_QDap · 2022-09-29
> > **Thanks**
> >
> > Thank you for your answer.
> >
> > I think I'm slightly understandting the rôle of $\theta$ in Project and Forget a bit better. In the quadratic case, say, $\theta$ somehow represents the (constant) dual error $C_{ij} - f_i - g_j$ (up to the $2\gamma$ factor), right? and for other dual regularizers, it is somewhat the same but for other kind of divergences; as in all cases, setting $\theta = 0$  enforces $\forall i,j,\ C_{ij} = f_i + g_j$.
> >
> > I'm still a bit struggling to understand how one "calculate $\theta$", and what comes first between "calculating $\theta$" and finding $f_i, g_j$. For instance, in the _mixed regularizers_ paragraph, it is said "calculating $\ŧheta$ is more difficult (...) $\theta is the root of $e^x + x + f_i + g_j - C_{ij}$", but initially, this quantity shall not be constant, right ?

---

> ### Author Response · Authors · 2022-10-11
> **Séjourné et al. 2019**
>
> We thank the reviewer for pointing us to this paper. We shall update the paper to add some discussion in relation to this work.
>
> We have now taken a look at the paper and can make the following comment. We agree with the reviewer that the work is very similar to ours. However, there are two key differences
>
> 1) They make their assumptions in relation to the primal problem. We make assumption in relation to the dual problem. The reviewer does point out that these might be equivalent. However, we do not think these are always equivalent. The two could be equivalent under some very general assumptions, but we feel this would require proof.
>
> 2) They adapt Sinkhorns to solve their problem. We use Project and Forget. They show linear convergence of Sinkhorn under some additional assumptions that (we think) are not needed for Project and Forget (which also has linear convergence).

---

### Review · Reviewer_vKfE · 2022-09-09

**Summary Of Contributions:**

The paper proposes a new regularized unbalanced optimal transport formulation (DROT, Problem 3). In contrast to many existing approaches to OT, the authors regularize the dual formulation of OT (which optimizes over potentials) instead of the primal (which conventionally optimizes over plans). Then the authors derive the respective regularized primal problem from the dual (Theorem 1).

The key regularization idea is to substract some strictly convex functions of the dual potentials from the standard duality formula. The main claim of the paper (bold in Section 1.1.1) is that this approach allows to easily control the level of mass creation and destruction by the proper choice of the dual regularizers. It is claimed that DROT approximates well the base Monge-Kantorovich problem, leads to sparse solutions and can be solved by the existing Project and forget algorithm.

**Broader Impact Concerns:**

No ethical considerations.

**Requested Changes:**

See my comments about the experiments and my other questions in the previous section.

**Strengths And Weaknesses:**

The detailed theoretical derivation and the analysis is the key strength. The authors carefully prove the (strong) duality, estimate the difference between the solution of their problem and the basic Monge-Kantotovich OT. They also prove that the support of their recovered plan is contained in that of the usual OT problem (for very high regularization parameter). Besides, they theoretically justify the behaviour of their method in terms of mass creation/destruction.

Nevertheless, I would argue that the paper does not provide enough experimental evidence to support their main claims. More precisely, I after looking through the experiments, as a reader, I am not sufficently convinced that this the proposed OT formulation is not "yet another OT", but something which indeed might be useful in ML problems. Specifically, the main feature which the authors propagate -- "easy mass destruction and creation" -- is not disclosed. I think the authors should deeper elaborate this feature. I details my concerns and questions about all this below.

(1) Figure 1 has unreadable legend and axis labels. From the text it is not very clear what does is actually plotted and how to interpret it and why it is related to sparsity. Please expand the explanation in the text.

(2) Figure 2 seems to be the experiment which is the most related to the main claim about the mass destruction and creation. However, it not clearly stated what is plotted in Figure 2 and the reader has to think out how exactly these plots are organized (for example, do you stack or overlay different bar plots?; why in the third plot the yellow bars are on top of everything?). This is not good for the presentation.

(3) The experiment of Section 4.2.2 (domain adaptation) does not provide any insight of the performance of the method. There is no special discussion why mass creation/destruction is useful for this particular problem. The authors just compare classification accuracy (after dataset transportation). Some versions of their method provide slightly better scores than the other methods, but I think this means nothing. Indeed, a recent paper [1] shows that the application of OT to unsepervised adaptation is questionable.

(4) Color transfer is a reasonable example but still a toy one. Besides, It is hard to judge the results of the transfer. Why do the authors not plot the underlying RGB color palletes of images? It would be nice to see how the mass movement behaves there (though might be a little bit hard to see the destruction/creation).

(5) The paper devotes limited attention to the practical computational aspect of DROT. I do not understand how scalable and applicable is the Project and Forget method applied to DROT, for example, compared to common Sinkhorn & Entropy based algorithms.

[1] Vacher, A., & Vialard, F. X. (2021). Convex transport potential selection with semi-dual criterion. arXiv preprint arXiv:2112.07275.

To conclude, the theoretical part of the paper is reasonable but the experimental evaluation is not sufficiently well conducted. As the main contribution of the paper (which the authors highlight a lot) is that their formulation allows to easily split/create the mass, I wonder in which ML application of OT we need to do so. I hope the authors could further develop the motivation and explain potential application prospects of their methodology. Currently, I see a reasonable theory which has (presumably) limited application perspectives in ML.

---

> ### Author Response · Authors · 2022-09-27
> **Response**
>
> We will increase the size of figure 1.
>
> The bar plots are overlaid. Hence we need to look at the top of the plot. For quadratic the top of the bars could be red or yellow hence saying both creation and destruction happens. For entropy it’s only red, showing creation. For Exponential it is only yellow showing destruction. We have made this plot bigger as well.
>
> We are not making any claims about the performance of the method. Just that the formulation has nice theoretical properties.
>
> The domain adaptation experiment is there to highlight the fact that creation versus destruction versus mixed produces different results.
>
> The method is slower than Sinkhorn, but as far as we know is state of the art for other formulations for which Sinkhorn doesn’t apply it is state of the art. Here is a table from the Project and Forget paper comparing the method for the quadratically regularized version.

---

> > ### Comment · Reviewer_vKfE · 2022-10-07
> > **Response**
> >
> > Dear authors, thanks for your answers and for updating the manuscript. However, it seems like my important comment about the motivation has not been adressed.
> >
> > Since this is a ML journal, as a reviewer, I expect to at least see a discussion (possibly supported by the relevant experiments, but notnecessary) about why this formulation may be useful for ML. More precisely, "when and why do we need to create/destroy the mass"? What are the intended uses of the proposed OT formulation? Could the authors please elaborate this question during the remaining time?
> >
> > I still think that none of the experiments of 4.2 answer the above-mentioned questions and there is no transparent discussion of this in the text. In particuar, the metrics in Table 1 seems nearly random and are not insightful at all what is the actual effect of mass creation/destuction. The answer of the authors "creation versus destruction versus mixed produces different results" is rather obvious and not explanatory. The experiment of 4.2.1 might potentially be insightful, but in my view, is not well developed. Thus, due to this, the overall paper feels a little bit incomplete (from ML prospects/relevance) and may require further improvement.

---

### Review · Reviewer_p6h9 · 2022-09-12

**Summary Of Contributions:**

This paper proposes a new formulation of optimal transport based on the regularization of the duals. Unlike more classical approaches  the authors consider a penalty on the duals but do not change the constraints corresponding to these duals. The authors show that for several types of regularization, the problem naturally involves mass construction/destruction. Thus the authors use this new formulation for the unbalanced optimal transport.


**Requested Changes:**

More details on these last points:

- Connections with classical OT unbalanced formulations:

I feel that there are missing connections that could be made with the work of (Blondel & al, 2018) by changing the formulation a bit. The authors show in theorem (1) that in the square case the proposed formulation is equivalent to that of (Blondel & al, 2018), which I think is very interesting. For the entropy case, the problem looks a bit more complicated and the authors consider exponential regularizers and add two more variables $c_1, c_2$ in order to satisfy the constraints (Remark, 1). I find this point a bit unfortunate, because these two variables are added in a somewhat artificial way and I have the impression that we can get away with it. If so, the proposed formulation would also be equivalent to that of (Blondel & al, 2018) for entropic regularizers.

My question is the following (but maybe I'm wrong): is it possible to get rid of these variables $c_1, c_2$ by considering a problem (4) with the convex conjugates $\phi^{*}$ instead of $\phi$ (same for $\varphi$)? Indeed the authors regularize the dual and it seems more natural to me to penalize with the conjugates rather than the functions themselves.

Moreover, I think we would fall on a problem (5) in theorem (1) but with $\phi, \varphi$ instead of conjugate, without needing to use $c_1, c_2$ variables for entropy. In this sense the proposed formulation would be exactly equivalent to that of (Blondel & al, 2018) but also for entropy. Moreover it would not change anything for the quadratic case because $\phi^{*} = \phi$ in this case.

- Algorithmic solution:

The second important point of my review is the algorithmic solution ''Project and Forget''. It is strictly based on another article Gilbert & Sonthalia (2020). I find that this solution is not sufficiently self-content because very few details are given to understand how this algorithm allows to solve the proposed problem and especially how it allows a solution "at large scales" (quote from the abstract).

The general description of the algorithm is as follows ''[Project and Forget] is a conversion of Bregman's cyclic method into an active set method and, as such, can solve large scale, highly constrained convex optimization problems.'' Here, it is difficult to understand exactly what this  ''Bregman's cyclic method''. I also doubt the ''large scale'' aspect stated by the authors. This claim is not justified by any experiments (e.g. runtimes and comparison with other methods).

- Clarity and writing:

The writing of the article and its clarity could be improved. I think that the introduction is a bit too generic and does not really present the motivations of this work. It is very quickly said ''There is a second class of formulations called unbalanced optimal transport'' but without explaining the interests of these formulations. Even if this is not a big problem in itself, I find that this is important to improve.

Other examples: I find that Proposition 2 is rather secondary, and that it breaks the flow of the article more than it illuminates the properties of the proposed formulation. There are a lot of technical definitions related to this proposition (def 2 & 3) that I don't find very useful to put in the main text. The proof in the main text doesn't make the reading easier either.

I think that this part should be moved to the supplementary, and that the result could be stated simply by keeping only the substance of the Proposition which says ''the dual potentials converge'' or by writing the conclusion presented by the authors ''The implication of the upper-hemicontinuity of the optimal policy correspondence is that any sequence of solutions ...''.  This could allow us to detail a little the algorithmic solution.

- Other unclear points:
  - In the proof of Theorem 1: "Due to the strict convexity and co-finiteness of we have that is a strictly convex function. and has a unique stationary point that corresponds to its global minimum." I think this point deserves more details. I don't understand exactly how co-finiteness comes into play here.
  - The convex conjugate is never properly defined.
  - What are these co-finite and Bregman functions? Are they much more restricted than the Bregman functions used in unbalanced transport? (i.e. can we characterize them a bit more precisely?). In other words: how generic is this formulation for regularizers other than quadratic/exponential?

- Experiments:

I find the experiments in domain and color transfer rather interesting. Some small remarks however: I find figures 1 and 2 not very readable and the legends should be enlarged. Moreover, the choice of DE/DQ for the names of the methods drowns out the authors' methods a bit. It might be wise to add a ''(ours)'' to underline the authors' method.

- Other comments:

  - The beginning of section 1.1 which talks about Monge's problem is rather clumsy (''The original version is that of Monge. The Monge problem howver has some drawbacks'')
  - I think "Problem 4" should be "Problem 3"

**Strengths And Weaknesses:**

- Pros:
  - I think the new formulation is elegant, and looks natural. The connection with the unbalanced transport is very interesting. The explanation between the penalized problem on the duals and the fact of creating or not mass is quite intriguing, and allows to enrich the understanding of the unbalanced transport.
  - Theorem 1 which makes the connection with more classical primal formulations brings a new interesting light

- Cons:
  - Several points are not clear: in particular some other connections with the work of (Blondel & al, 2018) seem important to me to highlight.
  - The algorithmic solution is not detailed enough, it is difficult to grasp the effectiveness of the approach in practice
  - The article can be improved from a writing and presentation point of view

---

> ### Author Response · Authors · 2022-09-27
> **Response**
>
> **Comparison to Blondel et al.**
> —
>
> What the reviewer suggests is interesting, however, we are not sure if when we compute the dual of that problem we would get back the problem that we currently have. This would require some work to see if it were true.
>
> **Project and Forget**
> —
> For efficacy please see the table in general response. Here the table from the Project and Forget paper shows that it outperforms other methods when solving the quadratically regularized problem.
>
> **Clarity**
> —
>
> We agree with the reviewer that moving the proof of proposition two to the appendix would help improve readability.
>
> We shall also add more details about why such functions have critical points. The general idea is if the function is differentiable, convex and coercive it has a critical point.
>
> We shall define convex conjugate in the main text.
>
> We do not have a good characterization. These are, however, Bregman functions and they are unbounded and superlinear (so can’t have linear growth) in certain directions. While this is a technical restriction, I think this is true for most functions we usually use.
>
> We shall increase the size of the 2 figures.

---

### Author Response · Authors · 2022-09-27
**General Response**

We thank the reviews for the detailed review and apologize for the delay in our response. Here we respond to the comments made by the reviewers. We would like to highlight that every part of the paper was found interesting by at least one of the reviewers.

There was a general comment about the effectiveness of Project and Forget. From the Project and Forget paper (https://arxiv.org/pdf/2005.03853.pdf), we show the following two tables. These tables are running times for solving the quadratically regularized version (that appears in Blondel et al.).

| Algorithm         | $501$                                 | $1001$                             | $5001$       | $10001$                            | 20001         |
|-------------------|---------------------------------------|------------------------------------|--------------|------------------------------------|---------------|
| Poject and Forget | 12                                    | 151                        | 1972         | 5909                       | 21665 |
| Cyclic Bregman    | 2681                                  | -                                  | -            | -                                  | -             |
| LBFGSB            | 24                                    | 162                                | 4080          | Ran out of memory |
| Mosek dual        | 56                                    | 328                                | 1927  | Ran out of memory |
| Mosek primal      | 5                             | Ran out of memory |
| CPLEX primal      | 105                                   | Ran out of memory |
| CPLEX dual         | Ran out of memory |
| PGD               | Ran out of memory |


Table shows the running time and table below shows the convergence statistics.

|                    | Objective                             |                 | Feasibility Error |             |
|--------------------|---------------------------------------|-----------------|-------------------|-------------|
| Solver             | Dual                                  | Primal          | Dual              | Primal      |
|                    | $n=501$          |
| Project and Forget | 3.8416077                             | 3.8416077       | 1.7e-09   | 0   |
| Mosek Dual         | 3.8414023                             | 3.8414023       | 3.8e-08           | 2.8e-10     |
| LBFGSB             | n/a                                   | 3.8416114       | n/a               | 0   |
| Mosek Primal       | 3.8303160                             | 3.8303203       | 3.5e-8            | 2.2e-11     |
| CPLEX Primal       | 3.8416376                             | 3.8416076       | 8.44e-07          | 1.13e-04    |
| CPLEX Dual         | Ran out of memory |
|                    | $n=1001$      |
| Project and Forget | 1.947532046                           | 1.947532046     | 2.0e-8    | 0   |
| Mosek Dual         | 1.947091229                           | 1.947091203     | 2.7e-08           | 8.4e-11     |
| LBFGSB             | n/a                                   | 1.947548404     | n/a               | 0   |
| Mosek Primal       | Ran out of memory |
| CPLEX Primal      | Ran out of memory |
| CPLEX Dual        | Ran out of memory |
|                    | $n=5001$         |
| Project and Forget | 3.946556152e-01                       | 3.946556154e-01 | 2e-08             | 0   |
| Mosek Dual         | 3.880175376e-01                       | 3.880175255e-01 | 1.2e-08   | 7.4e-11     |
| LBFGSB             | n/a                                   | 3.947709104e-01 | n/a               | 0   |
| Mosek Primal       | Ran out of memory |
| CPLEX Primal      | Ran out of memory |
| CPLEX Dual        | Ran out of memory |
|                    | $n=10001$       |
| Project and Forget | $0.197687348$                         | $0.197687348$   | 2.0e-8    | $0$ |
| Mosek Dual       | Ran out of memory |
| LBFGSB            | Ran out of memory |
| Mosek Primal      | Ran out of memory |
| CPLEX Primal      | Ran out of memory |
| CPLEX Dual        | Ran out of memory |
|                    | $n=20001$         |
| Project and Forget | $0.0989355070$                        | $0.0989355070$  |  2.0e-8    | $0$  |
| Mosek Dual        | Ran out of memory |
| LBFGSB             | Ran out of memory |
| Mosek Primal       | Ran out of memory |
| CPLEX Primal       | Ran out of memory |
| CPLEX Dual        | Ran out of memory |

---

> ### Comment · Reviewer_p6h9 · 2022-09-28
> **Answer to authors**
>
> Dear authors,
> Thank you for your response. From what the tables show, I'm a bit skeptical about the "large scale" aspect of the "Project and Forget" algorithm claimed by the authors several times in the article. For example, unless I missed something, $n = 5001$ implies $t \approx 32 \text{min}$, I wouldn't call that "large scale". Therefore, I think it is important to tone down the "large scale" statement a bit.
>
> More importantly, in my opinion, the article is not enough self-contained regarding this algorithm. I think it's important to add some more details in the main text about how this algorithm works, given that https://arxiv.org/pdf/2005.03853.pdf is fairly new, and 54 pages long. It is difficult, in this form, to understand what is done in practice.

---

> > ### Author Response · Authors · 2022-09-28
> > **Response**
> >
> > We shall tone down the large scale discussion and add in more details project and forget algorithmic description. We hope to update the pdf shortly.

---

### Author Response · Authors · 2022-09-29
**Updated PDF**

Based on reviewer comments we have updated the pdf in the following ways

1. Added more details about Project and Forget.
2. Fixed technical concerns by adding the assumption that $\phi, \psi$ are bounded. Also fixed typos pointed out in proofs.
3. Removed the claim large scale.
4. Increased the size of some of the figures.
5. Added other small details and fixed minor typos pointed out.

We however, have not updated the related works section to discuss the papers brought to our attention by the reviewers. We are still working on this. We hope to do this soon.

---

> ### Comment · Reviewer_p6h9 · 2022-09-29
> **Response**
>
> Dear authors, thank you. However I am not sure that the current pdf is changed compared to the previous one (e.g. there is still the "large-scale claim"). Could you highlight, maybe in a different color, when the changes are made ?

---

> > ### Comment · Reviewer_QDap · 2022-09-29
> > **I second this**
> >
> > I second comment by reviewer p6h9, highlighting the changes would clearly help the reviewing process to run smoother.

---

> ### Author Response · Authors · 2022-09-29
> **Apologies**
>
> Apologies, I uploaded the incorrect file before (so didnt have the changes). I uploaded the correct version this time. I think openreview has this feature for comparing the version that provides highlighted differences (revisions -> compare revisions -> select latest and first version -> at the bottom of the page). Please let me know if this is not the case, and I can manually highlight them.

---

### Decision · Action_Editors · 2022-10-18

**Recommendation:** Reject

**Comment:**

The paper proposes to perform regularization of the dual potentials in optimal
transport. The authors show that by choosing different regularizer on the dual, the
effect on the OT plan in the primal car create of destroy mass in the marginals
(similarly to Unbalanced OT). The authors then propose to use an efficient
"Project and Forget" algorithm to solve the resulting optimization problem. Then
they illustrate the effect of the regularizers on toy data (including image
colorization) and on a unique Domain Adaptation problem.

The paper was found unanimously interesting by the reviewers that also noted
several important questions (numerical large scale claim, relation with
unbalanced OT, unclear parts, lack of rigor in the mathematical results). The
authors provided some answers to some of the questions and changed the paper to
reflect them (smaller numerical claims, more details about project and forget,
some typos and mathematical rigor corrections). The authors claim that they
moved some of the proofs in appendix which does not appear in the revised
version.

As noted by some reviewers during the public and private discussions numerous
comments have not been addressed (or not enough) by the authors. The
overwhelming consensus among reviewers was to reject the paper because despite
its qualities (original idea, interesting interpretations of unbalanced
transport) it definitely needs some more work. For instance by taking into
account the reviews comments and by rewriting to make both the contribution and
the algorithm more clear for a Machine Learning audience. The authors are
strongly encouraged to take into account the reviewers comment before
resubmitting a major revision of the paper to TMLR.

A short summary of the necessary changes :
- The whole paper needs to be worked on to improve clarity and rigor in both the
  claims and theoretical justifications. Many problems were noted by the
  reviewers and not all of them have been taken into account in the edited
  version. All reviewers agreed that the paper is  hard to follow and that the
  proposed approach is not well motivated for an ML paper. The changes in the
  edited version were very small except some work on detailing the algorithm and
  did not reflect well all the comments from the reviewers.
- Positioning the DROT with Unbalanced OT (UOT) is necessary since it is the clear
  competitor of the method (when creation/destruction of mass is interesting).
  As discussed by reviewers the proposed method is probably equivalent to UOT
  for some choice of regularizer/divergence which is very interesting. The case
  where they are not equivalent should be discussed with examples of what
  DROT can do that UOT cannot. In addition to the necessary theoretical
  discussion  illustration would be to compare in the numerical experiment the
  creation/destruction of masses of the two approaches
- The algorithm Project and forget is interesting and seems to be a good fit for
  the problem. But it is not well introduced or explained. It has been proposed
  recently (and seems not to be published yet) which means that the reader
  cannot be expected to know it or go read the paper. As such the current paper
  is not self content. The authors did change the paper by explaining the
  algorithm with more details but it was deemed not sufficient by the reviewers
  that needed some clarifications in the discussion to better grasp it which is a
  clear indication that the writing is currently lacking.

**Audience:**

This paper and the use of this  new algorithm in this context can definitely be of
interest to a subset of the community. Still the lack of positioning of the method with
respect to classical Unbalanced OT (that also creates and destroy mass) greatly
limits the interest in the current version.


**Claims And Evidence:**

Some claims of the paper (theoretical) are well supported, other claims such as
the claim to solve large scale problems were removed form the paper due to lack of
evidence noted by reviewers.

But other important claims are not really supported. For instance the authors claim
that the formulation is novel but the reviewers all agree that there is a good
chance that the dual regularization is exactly equivalent to a well know
formulation called unbalanced OT in some conditions. This is an important questions that the authors
did not discuss in their replies or edited version.